# Chemogenetic attenuation of neuronal activity in the entorhinal cortex reduces Aβ and tau pathology in the hippocampus

Gustavo A. Rodriguez[1,2], Geoffrey M. Barrett[1,2], Karen E. Duff[1,2,3ᵒ]*, S. Abid Hussaini[1,2ᵒ]*

1 Taub Institute for Research on Alzheimer's disease and the Aging Brain, Columbia University Irving Medical Center, New York, New York, United States of America, 2 Department of Pathology and Cell Biology, Columbia University Irving Medical Center, New York, New York, United States of America, 3 UK Dementia Research Institute at University College London, London, United Kingdom

ᵒ These authors contributed equally to this work.
* sah2149@cumc.columbia.edu (SAH); k.duff@ucl.ac.uk (KED)

**Data Availability Statement:** The source data for all figures in this manuscript are available as supporting information in S1 Data. We have also

## Abstract

High levels of the amyloid-beta (Aβ) peptide have been shown to disrupt neuronal function and induce hyperexcitability, but it is unclear what effects Aβ-associated hyperexcitability may have on tauopathy pathogenesis or propagation in vivo. Using a novel transgenic mouse line to model the impact of human APP (hAPP)/Aβ accumulation on tauopathy in the entorhinal cortex–hippocampal (EC-HIPP) network, we demonstrate that hAPP overexpression aggravates EC-Tau aggregation and accelerates pathological tau spread into the hippocampus. In vivo recordings revealed a strong role for hAPP/Aβ, but not tau, in the emergence of EC neuronal hyperactivity and impaired theta rhythmicity. Chronic chemogenetic attenuation of EC neuronal hyperactivity led to reduced hAPP/Aβ accumulation and reduced pathological tau spread into downstream hippocampus. These data strongly support the hypothesis that in Alzheimer's disease (AD), Aβ-associated hyperactivity accelerates the progression of pathological tau along vulnerable neuronal circuits, and demonstrates the utility of chronic, neuromodulatory approaches in ameliorating AD pathology in vivo.

## Introduction

The accumulation of hyperphosphorylated, misfolded tau proteins into neurofibrillary tangles (NFT), coupled with deposition of amyloid-beta (Aβ) into extracellular plaques, are 2 hallmark pathological features of Alzheimer's disease (AD) in the brain. The severity of cortical NFT accumulation is strongly correlated with Aβ plaque load [1, 2] and is the principal neuropathological variable associated with cognitive impairment in AD [1–4]. The entorhinal cortex (EC; Brodmann Areas 28 and 34) is a structure in the parahippocampal gyrus that plays a critical role in spatial representation and navigation [5–7], and it is one of the first structures to exhibit AD-related tauopathy and subsequent neuronal loss [8, 9]. As AD progresses, considerable

made representative mouse position data during open field recording sessions available at the following Github repository: https://github.com/HussainiLab/PLOS-Biology-manuscript-data. Analysis tools such as BatchTintV3 and HfoGUI used in the manuscript are freely available on https://github.com/HussainiLab.

**Funding:** This work was supported by research grants from the National Institute on Aging https://www.nia.nih.gov (R01AG050425) to SAH and KED, (R01AG050425-Supplement) to SAH/GAR, the Alzheimer's Association https://www.alz.org (AARFD-17-504409) to GAR and the UK DRI to KED, which receives its funding from DRI Ltd, funded by the UK Medical Research Council, Alzheimer's Society and Alzheimer's Research UK. The funders had no role in study design, data collection and analysis, decision to publish, or preparation of the manuscript.

**Competing interests:** The authors have declared that no competing interests exist.

**Abbreviations:** AD, Alzheimer's disease; AIS, axon initial segment; APP, amyloid precursor protein; AT8, Ser202/Thr205 phosphotau-specific antibody; Aβ, amyloid-beta; BBB, blood–brain barrier; CaMKII$_\alpha$, calcium/calmodulin-dependent protein kinase type II subunit alpha; CA1, Cornu Ammonis 1; CNO, clozapine-n-oxide; CP27, human Tau130-150; DG, dentate gyrus; DREADD, designer receptor exclusively activated by a designer drug; EC, entorhinal cortex; eGFP, enhanced green fluorescent protein; FTD, frontotemporal dementia; hAPP, human amyloid precursor protein; hM3D$_q$, modified human M3 DREADD receptor; hM4D$_i$, modified human M4 DREADD receptor; HIPP, hippocampus; hTau, human tau; IHC, immunohistochemistry; ISF, interstitial fluid; LFP, local field potential; NFT, neurofibrillary tangle; ROI, region of interest; Sub, subiculum.

accumulation of pathological tau continues downstream into the hippocampus (HIPP), which is extensively connected to the EC. Preclinical investigation into the stereotypical spread of pathological tau along neuronal circuits in AD is an active area of research interest [10–13]. However, the biological mechanisms underlying the propagation of tau pathology in the brain are currently unresolved.

In vitro studies utilizing rodent primary neurons have provided several mechanistic insights into the pathophysiological relationship between cleaved amyloid precursor protein (APP) fragments and tau. At the cellular level, accumulating evidence implicates Aβ oligomers as a causative agent in the increased phosphorylation of tau at AD relevant epitopes [14] and the missorting of tau and neurofilaments within the cell [15]. In mouse models of tauopathy, stereotaxic injection of Aβ oligomers and fibrils into the brain results in significantly elevated phosphorylation of tau [16] and the increased induction of NFTs [17]. Thus, tauopathy in the brain may be aggravated by the increased production or accumulation of APP fragments in vivo through direct interaction. For reviews of experimental models that examine Aβ-induced tau alterations and pathology, see [18, 19].

Alternatively, human APP/Aβ accumulation in the brain may trigger the aggregation and acceleration of tau pathology via an intermediate, nonpathogenic mechanism. Indeed, several reports now describe an effect of Aβ accumulation on neuronal network hyperactivity in Aβ generating mouse models [20–22]; for review, see [23, 24], as well as in humans with mild cognitive impairment [25, 26]. Spontaneous, nonconvulsive epileptiform-like activity has been described in cortical and hippocampal networks of relatively young transgenic mice overexpressing mutant human amyloid precursor protein (hAPP) [21]. In addition, increased proportions of neurons surrounding amyloid plaques exhibit aberrant hyperactivity [20] and are accompanied by the breakdown of slow-wave oscillations [27]. Interestingly, neuronal hyperactivity has been shown to precede amyloid plaque formation in the hippocampus, suggesting that the abnormal accumulation of soluble Aβ drives aberrant neuronal network activity [28, 29]. Thus, it is plausible that Aβ-associated hyperactivity can facilitate the progression of pathological tau in the brain and does so indirectly without the need for direct Aβ-tau interaction. Interestingly, stimulating neuronal activity can facilitate both Aβ and tau release from neurons in vivo [30, 31] and exacerbates Aβ deposition and tauopathy in synaptically connected neurons [13, 32–34]. Mature tau pathology may in turn aggravate Aβ-associated neuronal network dysfunction by further altering neuronal firing rates and network oscillations [35], recruiting neuronal populations into a harmful feedback loop involving protein aggregation and aberrant signaling.

In these studies, we utilize a newly developed AD mouse line to resolve the individual effects of hAPP/Aβ and tau pathology on neuronal activity in the EC. Mice that generate Aβ and tau pathology were compared with littermates that generate either Aβ or tau pathology alone, while nontransgenic littermates served as controls. We first demonstrate that hAPP/Aβ aggravates tau accumulation in the EC and accelerates pathological tau spread into the HIPP, supporting previous findings [36–38]. In vivo electrophysiological recordings in our mice revealed distinct neuronal hyperactivity and network dysfunction associated with EC hAPP/Aβ expression but not tau expression. We then employed a chemogenetic approach in the transgenic mice to combat EC neuronal hyperactivity, with the goal of reducing the accumulation of pathological Aβ and tau along the entorhinal cortex–hippocampal (EC-HIPP) network. Chronic attenuation of EC neuronal activity dramatically reduced hAPP/Aβ accumulation in downstream hippocampus and reduced abnormally conformed and hyperphosphorylated tau aggregates along the EC-HIPP network. Our data support the emerging view that Aβ-associated hyperactivity plays a role in AD pathogenesis, specifically by acting as an accelerant of tau spread along synaptically connected neuronal circuits in the brain.

## Results

### Aβ-associated acceleration of tau pathology in EC-Tau/hAPP mice in vivo

Overexpression of mutant hAPP $_{(Swedish/Indiana)}$ in the hAPP/J20 mouse line leads to progressive Aβ plaque deposition throughout the hippocampus and neocortex, contributing to aberrant network activity and cognitive deficits [21, 22]. To test whether Aβ pathology alters the progression of tau pathology along the EC-HIPP circuit, we generated the hAPP/J20 x EC-Tau mouse line, hereafter referred to as EC-Tau/hAPP (described in Materials and methods). EC-Tau/hAPP mice sampled at 10, 16, and 23 months of age revealed a progressive, age-dependent accumulation of both hAPP/Aβ and tau pathology in the regions of interest (Fig 1A and 1B). Immunostaining for hAPP/Aβ using the anti-beta amyloid antibody 6E10 revealed appreciable extracellular Aβ accumulation in the hippocampus and EC at 16 and 23 months compared with 10 months of age. Diffuse amyloid accumulation made up the majority of the pathology, though small, compact Aβ plaques and large, dense-core Aβ plaques were also present at 16 months (Fig 1A). At 10 months, immunostaining for abnormally conformed, misfolded tau (MC1 antibody) revealed mostly diffuse tau in neuropil throughout the EC and in axons terminating in the middle- and outer-molecular layers of the dentate gyrus (DG) (Fig 1B). No somatodendritic MC1+ staining was detected in hippocampal subregions at this age. By 16 months, it was clear that tau aggregation had not only increased in the EC but had also begun to appear in hippocampal subregions, including the DG, Cornu Ammonis 1 (CA1), and subiculum (Sub). This accumulation of misfolded tau in hippocampal neurons was notably heightened at 23 months of age.

To determine whether hAPP/Aβ pathology affects pathological tau accumulation within the hippocampus, we compared 16-month EC-Tau/hAPP mice ($n = 6$) to age-matched littermates that express mutant human tau (hTau) alone (EC-Tau, $n = 5$). At 16 months, EC-Tau/hAPP mice exhibit robust Aβ deposition throughout the EC-HIPP, whereas EC-Tau mice do not exhibit any 6E10+ immunoreactivity (Fig 1C and 1D). MC1+ immunostaining revealed a dramatic increase in abnormal, misfolded tau within the somatodendritic compartments of EC and HIPP neurons of EC-Tau/hAPP mice compared with EC-Tau littermates (Fig 1E and 1F). This increase was nearly 4-fold higher in EC cells and over 20-fold higher in DG granule cells (Fig 1G and 1H), suggesting that hAPP/Aβ expression in the EC-HIPP network accelerates tau pathology along the classical perforant pathway. Thus, we chose to examine EC neuronal activity patterns in 16-month EC-Tau/hAPP mice to determine whether hAPP/Aβ was negatively impacting the local neuronal population and driving tau pathology.

### Aβ-associated EC hyperactivity and network dysfunction

In vivo multielectrode recordings were performed in the EC of 16-month EC-Tau/hAPP mice and their age-matched, transgenic, and nontransgenic littermates, as previously described [39]. Briefly, tetrodes from custom-built microdrives (Axona, UK) were positioned to target cell layers II/III in the dorsal EC (see Materials and methods). Single-unit activity and local field potentials (LFPs) were collected while mice freely explored a large open field arena. Each experimental mouse underwent 4 to 6 recording sessions total over the course of several days, with only one recording session performed per day. Tetrodes for each mouse were moved down 0.1 mm from their previous position 24 hours prior to the next recording session, allowing stable electrode positioning and a robust sampling of EC LII/III neuronal activity for each mouse. Immunohistochemistry (IHC) was performed on post-mortem brain sections at the end of the study to confirm electrode placement (Fig 2A) and to examine the distribution of Aβ and tau pathology in the EC-HIPP network.

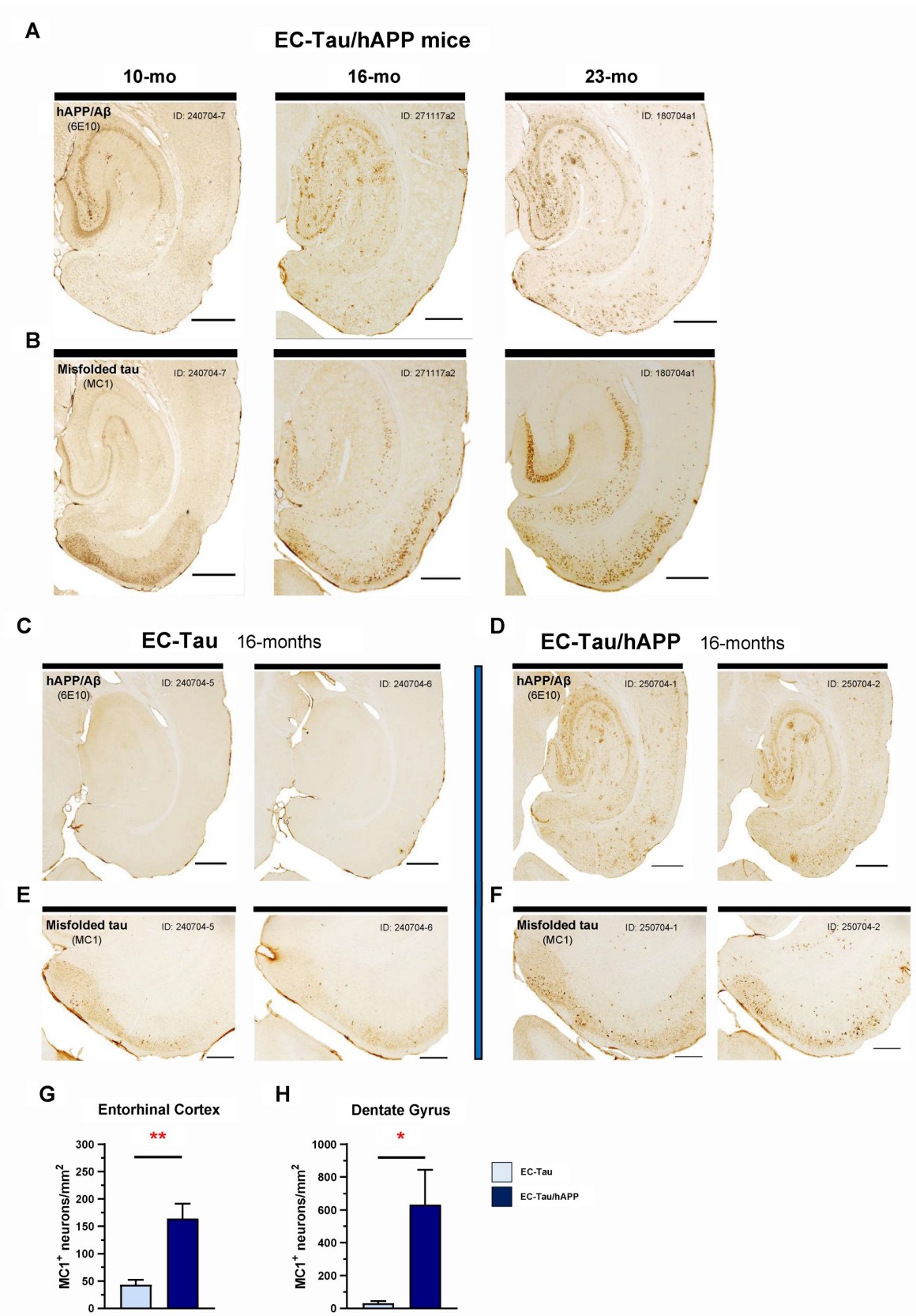

**Fig 1. hAPP/Aβ-associated acceleration of tau pathology along the EC-HIPP network.** The EC-Tau/hAPP mouse line was created to model functional interactions between hAPP/Aβ and hTau pathologies in a well-characterized neuronal circuit in vivo (EC-HIPP network). ***A-B.*** Horizontal brain sections from a sample of 10-, 16-, and 23-month-old EC-Tau/hAPP mice were processed for immunohistochemical detection of hAPP/Aβ (6E10, anti-beta amyloid) and abnormal, misfolded tau (MC1, conformationally dependent). A clear, age-dependent progression of Aβ and tau pathology within the EC-HIPP network was evident. Importantly, MC1+ immunostaining revealed increased pathological tau within hippocampal neurons at 16 months compared with 10 months of age. Scale bars, 500 μm. ***C-D.*** To determine whether hAPP/Aβ aggravates tau accumulation and accelerates pathological tau spread at the 16-month time point, horizontal brain sections from EC-Tau/hAPP mice ($n = 6$ total; $n = 3$ female, $n = 3$ male) and age-matched EC-Tau littermates ($n = 5$ total: $n = 3$ female, $n = 2$ male) were processed for 6E10+ and MC1+ immunostaining. Representative, adjacent brain sections from 2 mice sampled are shown for both 6E10 and MC1. The 16-month EC-Tau/hAPP mice exhibited robust Aβ accumulation and plaque deposition throughout the EC and HIPP. Diffuse Aβ accumulation comprised the majority of the pathology in these regions, with occasional small, compact plaques and large, dense-core Aβ plaques present. EC-Tau mice did not exhibit 6E10+ immunoreactivity. Scale bars, 500 μm. ***E-F.*** MC1+ immunostaining revealed an acceleration of tau pathology in the EC of EC-Tau/hAPP mice, characterized by an increased number of neurons with abnormally conformed tau localized within somatodendritic compartments. Scale bars, 250 μm. ***G-H.*** Semiquantitative analysis of MC1+ cell counts was performed in the EC and DG of both EC-Tau/hAPP and EC-Tau mice. Mean MC1+ cell counts (MC1+ neurons/mm$^2$) in EC-Tau/hAPP brain sections were over 4-fold and over 20-fold greater than EC-Tau brain sections in the EC and DG, respectively. EC: EC-Tau/hAPP, 164.02 ± 27.12 versus EC-Tau, 43.23 ± 9.35. DG: EC-Tau/hAPP, 631.59 ± 212.90 versus EC-Tau, 30.95 ± 14.20. Unpaired *t*-tests with Welch's correction: EC, $t = 4.211$, $p < 0.01$; DG, $t = 2.815$, $p < 0.05$. Graphs and numerical values in figure legend represent mean ± SEM for the averaged MC1+ neurons/mm$^2$ values from 3 independently processed brain sections per mouse. $^*p < 0.05$; $^{**}p < 0.01$. Source data are available in S1 Data. Aβ, amyloid-beta; DG, dentate gyrus; EC, entorhinal cortex; hAPP, human amyloid precursor protein; HIPP, hippocampus; hTau, human tau; MC1, antibody for misfolded tau; SEM, standard error mean.

A total of 1,910 EC neurons were recorded and analyzed from 31 mice (control, $n = 8$; EC-Tau, $n = 7$; hAPP, $n = 8$; EC-Tau/hAPP, $n = 8$) (Fig 2) (for detailed methodology, see Materials and methods). Plotting the cumulative frequency distributions of the spontaneous, average firing rates of all recorded neurons showed a clear shift toward higher firing rates in both EC-Tau/hAPP mice and hAPP mice versus control mice (Fig 2B). Individual 2-sample Kolmogorov–Smirnov tests comparing the distributions of EC-Tau/hAPP and hAPP neuronal firing rates to controls confirmed this shift ($p < 0.001$). An average firing rate was calculated per mouse from individual EC neurons, and then mean values per group were compared across genotype (Fig 2B, insert). EC-Tau/hAPP and hAPP mice exhibited nearly a 2-fold increase in their average firing rates versus control mice ($p < 0.001$, Dunnett's multiple comparisons test). Interestingly, the average firing rates of EC-Tau mice were not significantly different from controls ($p > 0.05$), suggesting that the increased hyperactivity in EC-Tau/hAPP neurons was driven by hAPP expression and/or Aβ deposition but not tau. To examine task-relevant neuronal firing rates during active exploration, we applied a minimum–maximum speed filter (5–100 cm/second) to the data to remove EC spiking activity occurring during bouts of immobility (S1A Fig). EC-Tau/hAPP and hAPP mice once again exhibited significantly increased average firing rates compared with control mice in speed-filtered datasets ($p < 0.01$). We then examined the interspike interval (ISI) values for EC single units per mouse, which served as an additional metric to verify neuronal hyperactivity within our dataset. Cumulative frequency distributions of the median ISIs showed that EC neurons from EC-Tau/hAPP and hAPP mice were skewed toward shorter ISIs than controls ($p < 0.001$, 2-sample Kolmogorov–Smirnov tests) (Fig 2C), which was confirmed by performing a Kruskal–Wallis test and Dunn's post hoc test (EC-Tau/hAPP versus control: $p < 0.05$; hAPP versus control: $p < 0.01$) (Fig 2C, insert). Interestingly, a 2-sample Kolmogorov–Smirnov test identified a subtle shift in the distribution of median ISIs from EC-Tau neurons versus control ($p = 0.039$), though median ISI values were not significantly different after post hoc analysis ($p > 0.05$).

Interneuron dysfunction has previously been described in hAPP-expressing mice, suggesting that shifts in the excitation–inhibition ratio within cortical and hippocampal networks may be responsible for emergent epileptiform activity [21, 40]. To examine cell-type-specific firing patterns in our single-unit dataset, we plotted cell frequency as a function of the neuronal waveform's averaged spike width (μs) (Fig 2D) [39, 41]. This resulted in a clear bimodal

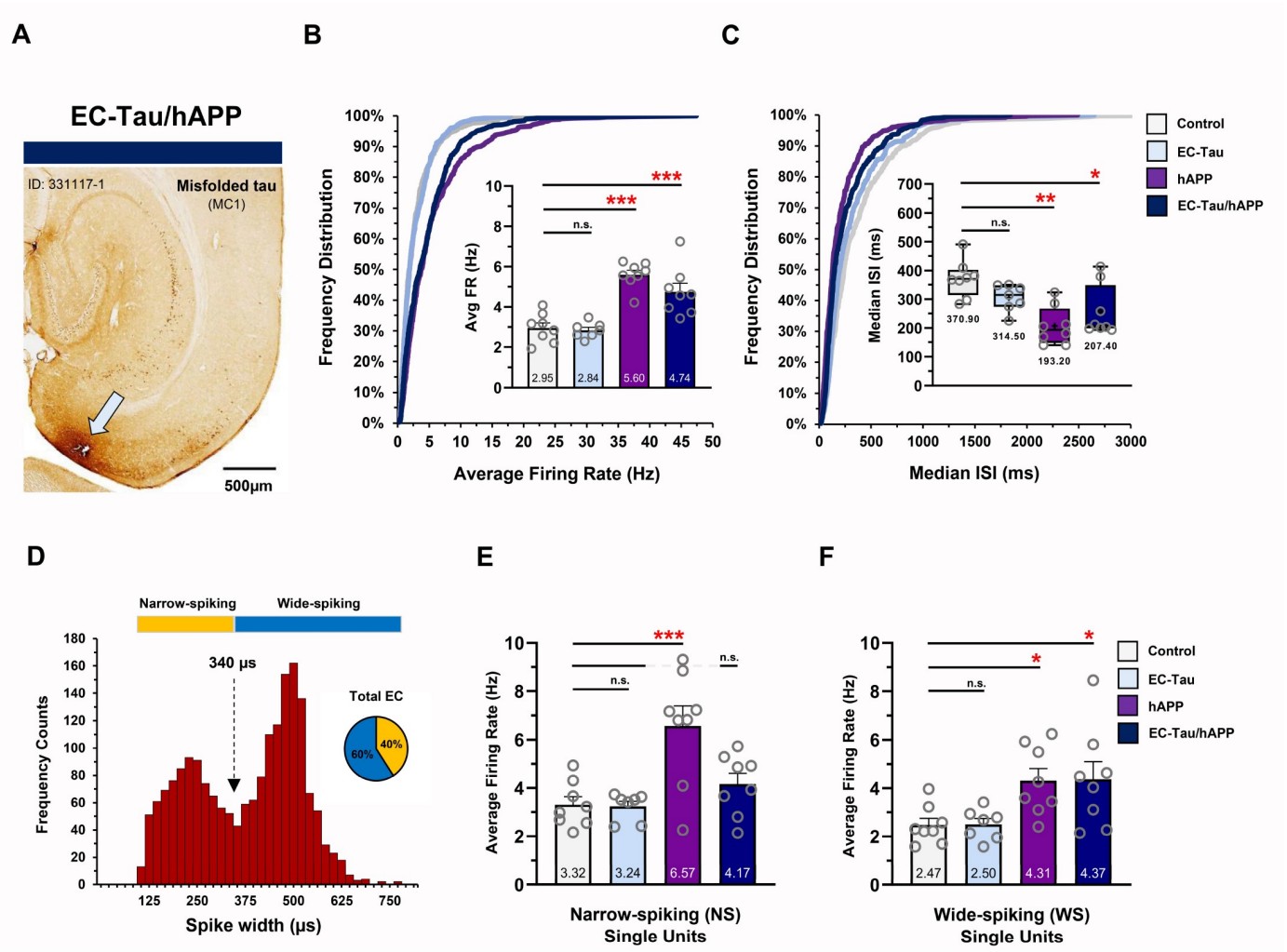

**Fig 2. Aβ-associated hyperactivity in EC single units in vivo.** In vivo multielectrode recordings were performed in the EC of 16-month EC-Tau/hAPP mice ($n = 8$) and their age-matched littermates: EC-Tau ($n = 7$), hAPP ($n = 8$), and nontransgenic control mice ($n = 8$). *A.* MC1[+] horizontal brain section from an EC-Tau/hAPP mouse depicting tetrode tract mark (arrow). Scale bar, 500 μm. *B.* Cumulative frequency distributions of the neuronal average firing rates for all single units collected. Neurons from EC-Tau/hAPP and hAPP mice exhibit distribution shifts toward higher firing rates versus control. Two-sample Kolmogorov–Smirnov test: EC-Tau/hAPP versus control: D = 0.263, $p < 0.001$; hAPP versus control: D = 0.281, $p < 0.001$. Insert, average firing rates for EC-Tau/hAPP and hAPP mice were nearly 2-fold higher than Control. One-way ANOVA: $p < 0.001$. *C.* Cumulative frequency distributions of EC neuron median ISIs show shifts in the distributions of both EC-Tau/hAPP and hAPP mice versus control. Two-sample Kolmogorov–Smirnov test: EC-Tau/hAPP versus control: D = 0.217, $p < 0.001$; hAPP versus control: D = 0.284, $p < 0.001$. Insert, box-and-whisker plots depicting median and mean (+) ISI values per genotype. Median ISI values were significantly decreased in EC-Tau/hAPP and hAPP mice compared with control. Kruskal–Wallis test: $p < 0.01$. *D.* Neuronal firing rates were binned and plotted as a function of waveform spike width. The resulting frequency histogram showed a bimodal distribution. A cutoff of 340-μm spike width was used to delineate putative interneurons (NS cells; yellow) from putative excitatory neurons (WS, cells; blue) (broken arrow). *E.* hAPP mice exhibit increased NS cell firing rates versus control. One-way ANOVA: $p < 0.01$. *F.* EC-Tau/hAPP and hAPP mice exhibit increased WS cell firing rates versus Control. One-way ANOVA: $p < 0.001$. Bar graphs represent mean ± SEM. Box-and-whisker plots represent median with minimum thru maximum values shown. Individual values per mouse appear as overlays. For Animal ID and sex of individual mice, please see S1 Data. $^*p < 0.05$; $^{**}p < 0.01$; $^{***}p < 0.001$. Source data are available in S1 Data. Aβ, amyloid-beta; DG, dentate gyrus; EC, entorhinal cortex; hAPP, human amyloid precursor protein; HIPP, hippocampus; ID, identification; ISI, interspike interval; MC1, antibody for misfolded tau; NS, narrow-spiking; SEM, standard error mean; WS, wide-spiking.

distribution of the binned cell population, wherein narrow-spiking (NS) cells could be separated from wide-spiking (WS) cells (Fig 2D–2G). Previous studies indicate that NS cells and WS cells likely correspond to putative interneurons and putative excitatory cells, respectively [41]. A total of 780 NS cells and 1,130 WS cells were identified from our EC recordings across 31 mice (NS, 40.84% versus WS, 59.16% of total neurons) (for detailed per animal-cell-type

information, see S1 Data and Materials and methods). Interestingly, hAPP mice exhibited increased neuronal firing rates in NS cells compared with control mice ($p < 0.001$), whereas EC-Tau/hAPP mice and EC-Tau mice did not ($p > 0.05$) (Fig 2E). In WS cells, both EC-Tau/hAPP mice and hAPP mice exhibited increased firing rates versus control ($p < 0.05$) (Fig 2F). EC-Tau mice did not exhibit altered WS firing rates compared with control ($p > 0.05$). Applying a speed filter to the dataset revealed similar effects on EC firing rates (S1B and S1C Fig). These data agree with previous reports describing Aβ-associated dysfunction in hippocampal interneurons and expand this hyperactivity phenotype to both putative interneurons and putative excitatory neurons in EC.

Impaired neuronal network activity has been described in human AD [42] and in mouse models of AD pathology [43–45]. To investigate the effects of hAPP/Aβ and tau pathology on EC network activity in vivo, we examined the LFPs and compared oscillatory activity across genotypes (Fig 3). Initial visual inspection of the filtered LFP signal in theta (4–12 Hz), low-gamma (35–55 Hz), and high-gamma (65–120 Hz) frequency ranges, as well as the filtered LFP spectrograms, revealed impaired theta rhythmicity in 16-month EC-Tau/hAPP mice compared with control mice (Fig 3A and 3B). Average percentage theta power was significantly decreased in EC-Tau/hAPP mice ($p < 0.01$) and hAPP mice ($p < 0.05$) versus control. Theta power was not significantly affected in EC-Tau mice ($p > 0.05$), and no genotype differences were detected in the low-gamma or high-gamma frequency ranges. Because running speed can impact theta power in the hippocampal formation [46, 47], we applied a minimum percentage maximum speed filter (5–30 cm/second) to the LFP recordings and reanalyzed the data to remove activity during bouts of immobility (S1D–S1F Fig). Similarly, we found reduced percentage theta power in EC-Tau/hAPP mice versus control ($p < 0.05$), although this was accompanied by an increase in percentage low-gamma power that was associated with hAPP/Aβ expression. Averaged, speed-filtered percentage high-gamma power values were not significantly different across genotypes.

Overexpression of hAPP$_{Swedish/Indiana}$ has previously been associated with increased locomotor activity in the hAPP/J20 mouse line [22, 48, 49]. To investigate locomotor activity in 16-month EC-Tau/hAPP mice, we analyzed the position data from our recorded mice as they performed a foraging task in the open field arena (S2 Fig). We did not detect significant differences between groups in the total distance traveled in the arena (m), percentage of arena coverage or average speed (cm/second) during exploration ($p > 0.05$) (S2A and S2B Fig). To examine whether group differences in locomotor activity were present within the initial phases of recording, we split the sessions and examined performance measures in the first 5, 10, and 15 minutes (S2C Fig). No significant differences were detected between groups in any measure and in any time-bin examined ($p > 0.05$), suggesting that motivated foraging behavior was not affected by Aβ and tau pathology in the open field arena.

Taken together, our in vivo recording data in the EC-Tau/hAPP mouse line suggest that at 16 months, hAPP/Aβ expression is associated with EC neuronal hyperactivity in both putative interneurons and excitatory neurons and that tau expression may selectively blunt putative interneuron hyperactivity in EC-Tau/hAPP mice. hAPP/Aβ expression is also associated with a distinct decrease in percentage theta power in both speed-filtered and unfiltered EC LFPs. Finally, tau pathology does not appear to overtly impact EC neuronal activity at 16 months.

## Chemogenetic attenuation of EC neuronal activity in vivo reduces Aβ accumulation in downstream hippocampus

Previous reports have shown that elevating neuronal activity can increase Aβ deposition and tau pathology in vivo [32–34]. However, these effects were restricted to mouse lines that

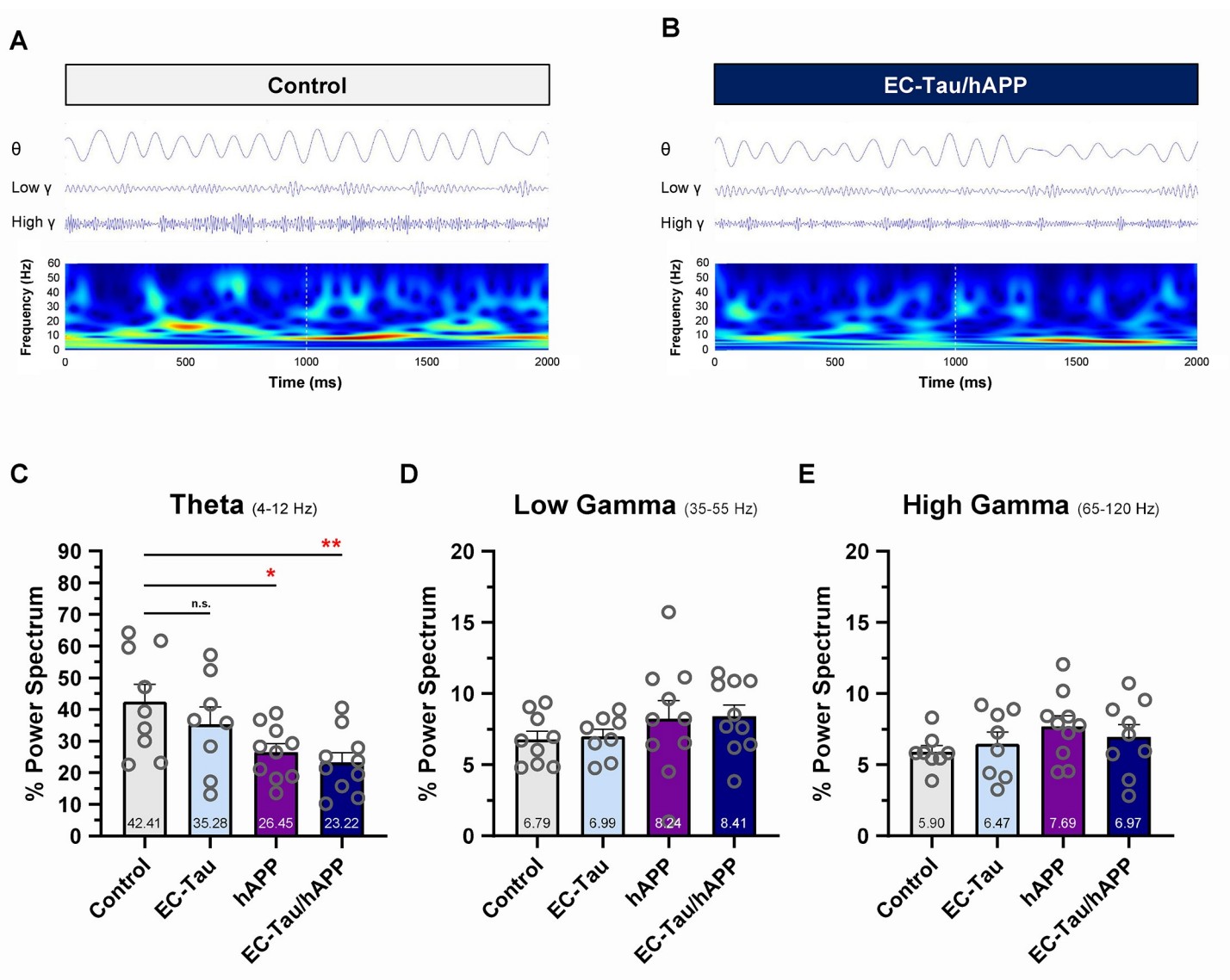

**Fig 3. Aβ-associated impairment of EC network activity in vivo.** LFPs were collected within the EC of 16-month EC-Tau/hAPP (*n* = 10 total: *n* = 6 female, *n* = 4 male) mice and age-matched littermates (hAPP, *n* = 10 total: *n* = 5 female, *n* = 5 male; EC-Tau, *n* = 8 total: *n* = 4 female, *n* = 4 male; nontransgenic control, *n* = 9 total: *n* = 5 female, *n* = 4 male). The percentage power values for oscillatory frequency bands were then calculated and compared across genotype. ***A-B.*** Representative, filtered LFP waveforms in the theta (4–12 Hz), low-gamma (35–55 Hz), and high-gamma (65–120 Hz) frequency ranges are shown for 1 control mouse and 1 EC-Tau/hAPP mouse along with the LFP spectrograms. LFP signals show disturbances in the theta rhythm of EC-Tau/hAPP mice. ***C.*** EC-Tau/hAPP mice and hAPP mice exhibited a significant decrease in percentage theta power versus control. EC-Tau mice were not significantly different from control. One-way ANOVA test: *p* < 0.05. Dunnett's multiple comparisons test: EC-Tau/hAPP versus control, *p* < 0.01; hAPP versus control, *p* < 0.05. ***D-E.*** No significant differences were detected across genotype when examining percentage power spectrum values in the low-gamma or high-gamma frequency ranges. One-way ANOVA: *p* > 0.05. Bar graphs represent mean ± SEM. Individual values per mouse appear as overlays. *\*p* < 0.05; *\*\*p* < 0.01. Source data are available in S1 Data. Aβ, amyloid-beta; EC, entorhinal cortex; hAPP, human amyloid precursor protein; LFP, local field potential; SEM, standard error mean.

individually overexpress hAPP and deposit Aβ/amyloid or overexpress mutant hTau and accumulate pathological tau in the brain. In vivo studies that examine the synergistic effects of Aβ-associated neuronal hyperactivity on tau pathology are necessary, as are experiments that test whether direct neuromodulatory intervention at the site of hyperactivity can impact the acceleration of tau pathology along neuronal networks. Therefore, we employed a chemogenetic approach in a series of experiments wherein a modified form of the human muscarinic M4

receptor (hM4D$_i$) was virally delivered and expressed in the EC of 16-month EC-Tau/hAPP mice and activated for 6 weeks. Activation of hM4D$_i$ designer receptors exclusively activated by a designer drug (DREADDs) via clozapine-n-oxide (CNO) engages the G$_i$ signaling pathway and effectively reduces neuronal firing rates in transduced cell populations [50, 51].

Single, high-dose CNO (5 and 10 mg/kg, intraperitoneal) injections reliably induced DREADDs activation and altered EC neuronal activity (S3A and S3B Fig) and percentage theta power (S3C and S3D Fig), providing valuable metrics for detecting chronic DREADDs activation in vivo. This was demonstrated in 16-month EC-Tau/hAPP mice ($n = 8$) implanted with osmotic minipumps loaded with CNO and expressing hM4D$_i$ DREADDs targeted to EC excitatory neurons. hM4D$_i$ EC DREADDs receptors were activated for 6 weeks by continuous delivery of CNO (1mg kg$^{-1}$ day$^{-1}$) into the peritoneum. Percentage theta power was reliably decreased in vivo by the last recording session of the treatment regimen (week 6 versus baseline, $p < 0.05$) (Fig 4A), whereas EC spiking activity was decreased at week 5 ($p < 0.05$) and week 6 ($p < 0.001$) (Fig 4B). Automated spike sorting and manual cluster cutting was then performed on recording data collected at baseline and week 6, resulting in a total of 285 EC neurons across CNO-treated EC-Tau/hAPP mice (baseline, $n = 154$ neurons; week 6, $n = 131$ neurons) (Fig 4C). We found that chronic hM4D$_i$ EC DREADDs activation decreased the average firing rate of EC neurons at week 6 versus baseline ($p < 0.01$). Further analysis by neuronal subtype revealed decreased firing rates in NS ($p < 0.05$) and WS ($p < 0.05$) cells at week 6 (Fig 4D and 4E). These data confirm that chronic CNO treatment decreased neuronal network and total spiking activity in our mice, as well as neuronal firing rates through activation of hM4D$_i$ EC DREADDs in vivo.

After 6 weeks of hM4D$_i$ EC DREADDs activation, all animals were euthanized, and immunostaining was performed on horizontal brain sections to confirm EC DREADDs expression and electrode placement and to identify pathological Aβ and/or tau deposition along the EC-HIPP network. An overlay of mCherry signal (hM4D$_i$ DREADDs) and enhanced green fluorescent protein (eGFP, control virus) expression patterns for experimental EC-Tau/hAPP mice is shown (Fig 5A). hM4D$_i$ DREADDs expression in the right hemisphere was primarily localized to cell bodies and neuropil throughout the EC, pre- and parasubiculum and Sub. Importantly, we did not detect mCherry signal (hM4D$_i$ DREADDs) crossover into the contralateral left hemisphere. EC-Tau/hAPP mice ($n = 9$) that were administered 1mg kg$^{-1}$ day$^{-1}$ CNO for 6 weeks exhibited reduced hAPP/Aβ accumulation within the right hippocampus, downstream of the DREADDs-activated right EC (Paired $t$-test: $t(8) = 5.919$, $p < 0.001$) (Fig 5B and 5C). Chronic EC DREADDs activation appeared to mainly impact diffuse Aβ accumulation rather than medium to large Aβ deposits, which were present in all HIPP subregions. Hippocampal 6E10+ immunoreactivity was not significantly different in right-versus-left hemispheres of 16–18-month mice under control conditions (Paired $t$-test: $t(8) = 1.357$, $p > 0.05$) (Fig 5D). No significant differences were detected between right-versus-left hippocampal region of interest (ROI) area (mm$^2$) sampled for 6E10+ immunostaining comparisons (S4A–S4C Fig).

These data provide evidence to support the utility of chronic, neuromodulatory intervention in the EC-HIPP network of hAPP/Aβ-expressing mice, since 6E10+ immunoreactivity was significantly reduced in the downstream HIPP after 6 weeks of hM4D$_i$ EC DREADDs activation.

## Chemogenetic attenuation of EC neuronal activity in vivo reduces tau pathology in downstream hippocampal subregions

We have previously shown that increased neuronal activity can aggravate tau pathology in EC-Tau mice [32]. Similarly, chemogenetic activation of neuronal activity in hTau mice

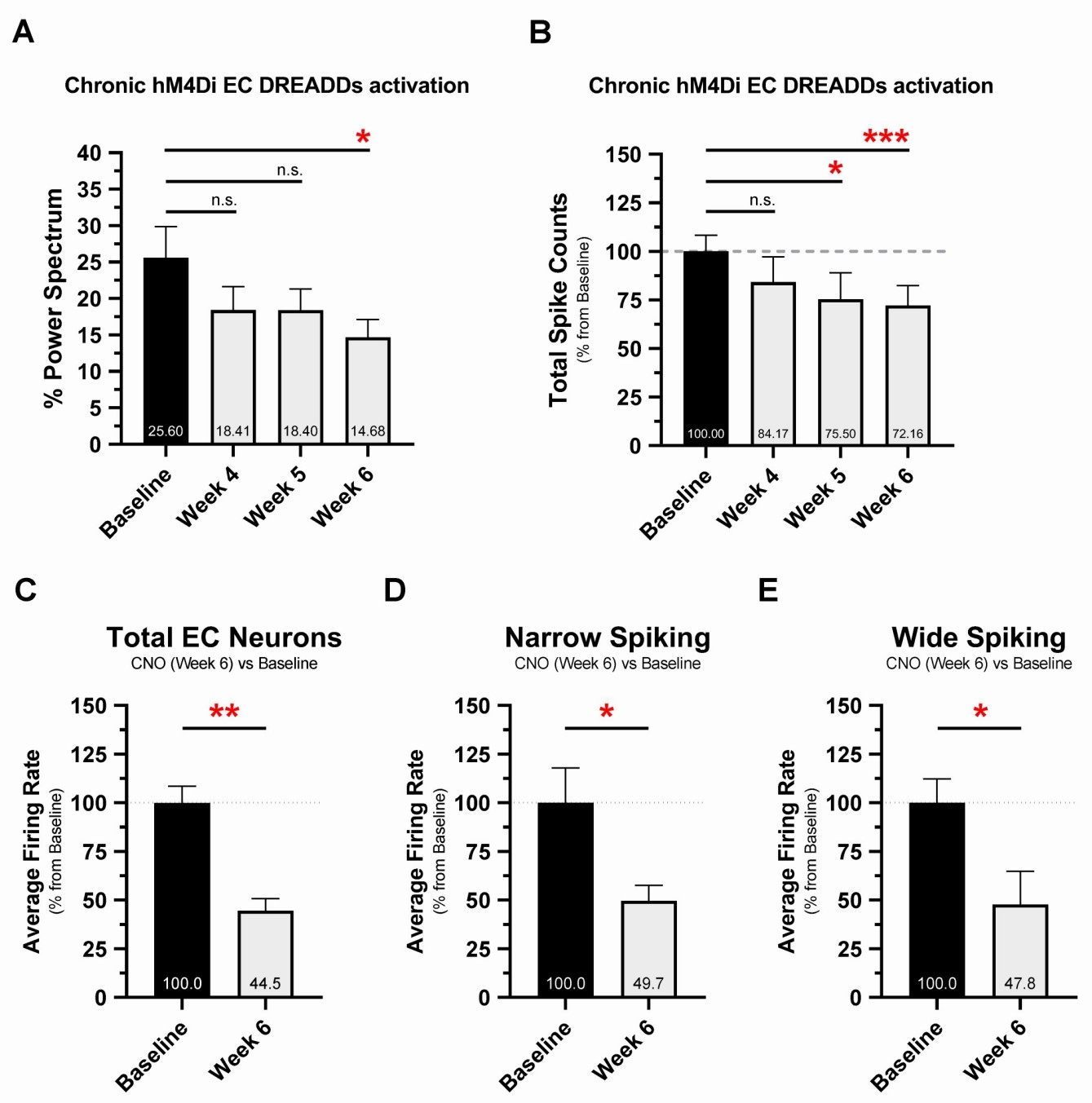

**Fig 4. Chronic hM4D$_i$ EC DREADDs activation reduces EC neuronal network activity and single-unit firing rates at 6 weeks.** The 16-month EC-Tau/hAPP mice were subjected to 6 weeks of DREADDs activation via osmotic minipump (CNO, 1 mg kg$^{-1}$ day$^{-1}$). In vivo neuronal activity was examined at 4, 5, and 6 weeks of CNO treatment and compared with baseline activity (presurgery) measures. **A.** Chronic hM4D$_i$ EC DREADDs activation reduced percentage theta power in EC-Tau/hAPP mice ($n$ = 8 total: $n$ = 4 female, $n$ = 4 male). Repeated-measures ANOVA: $F_{(7,21)}$ = 4.697, $p < 0.05$. Dunnett's multiple comparisons test: week 4 versus baseline, $p > 0.05$; week 5 versus baseline, $p > 0.05$; week 6 versus baseline, $p < 0.05$. **B.** Chronic hM4D$_i$ EC DREADDs activation also reduced total spiking activity in EC-Tau/hAPP mice. Repeated-measures ANOVA: $F_{(7,21)}$ = 11.270, $p < 0.001$. Dunnett's multiple comparisons test: week 4 versus baseline, $p > 0.05$; week 5 versus baseline, $p < 0.05$; week 6 versus baseline, $p < 0.001$. **C.** Automated spike sorting and manual cluster cutting was performed on neuronal spiking data from 2 recording sessions per EC-Tau/hAPP mouse (baseline and week 6, CNO). Chronic hM4D$_i$ EC DREADDs activation reduced Total EC neuron average firing rates at week 6 versus baseline. Paired $t$-test: $t$ (6) = 4.972, $p = 0.0025$. **D-E.** Single units were then categorized as narrow-spiking or wide-spiking according to spike width. Average firing rates were reduced at week 6 versus baseline in both narrow-spiking EC neurons (Paired $t$-test: $t$ (6) = 2.635, $p = 0.0388$) and wide-spiking EC neurons (Paired $t$-test: $t$ (6) = 2.804, $p = 0.0310$). Graphical representations appear as mean ± SEM. Normalized data are shown as a percentage of baseline. $^{*}p < 0.05$, $^{**}p < 0.01$. $^{***}p < 0.001$. Source data are available in S1 Data. Aβ, amyloid-beta; CNO,

clozapine-n-oxide; DREADD, designer receptor exclusively activated by a designer drug; EC, entorhinal cortex; hAPP, human amyloid precursor protein; hM4D$_i$, human M4 DREADD receptor; LFP, local field potential; SEM, standard error mean.

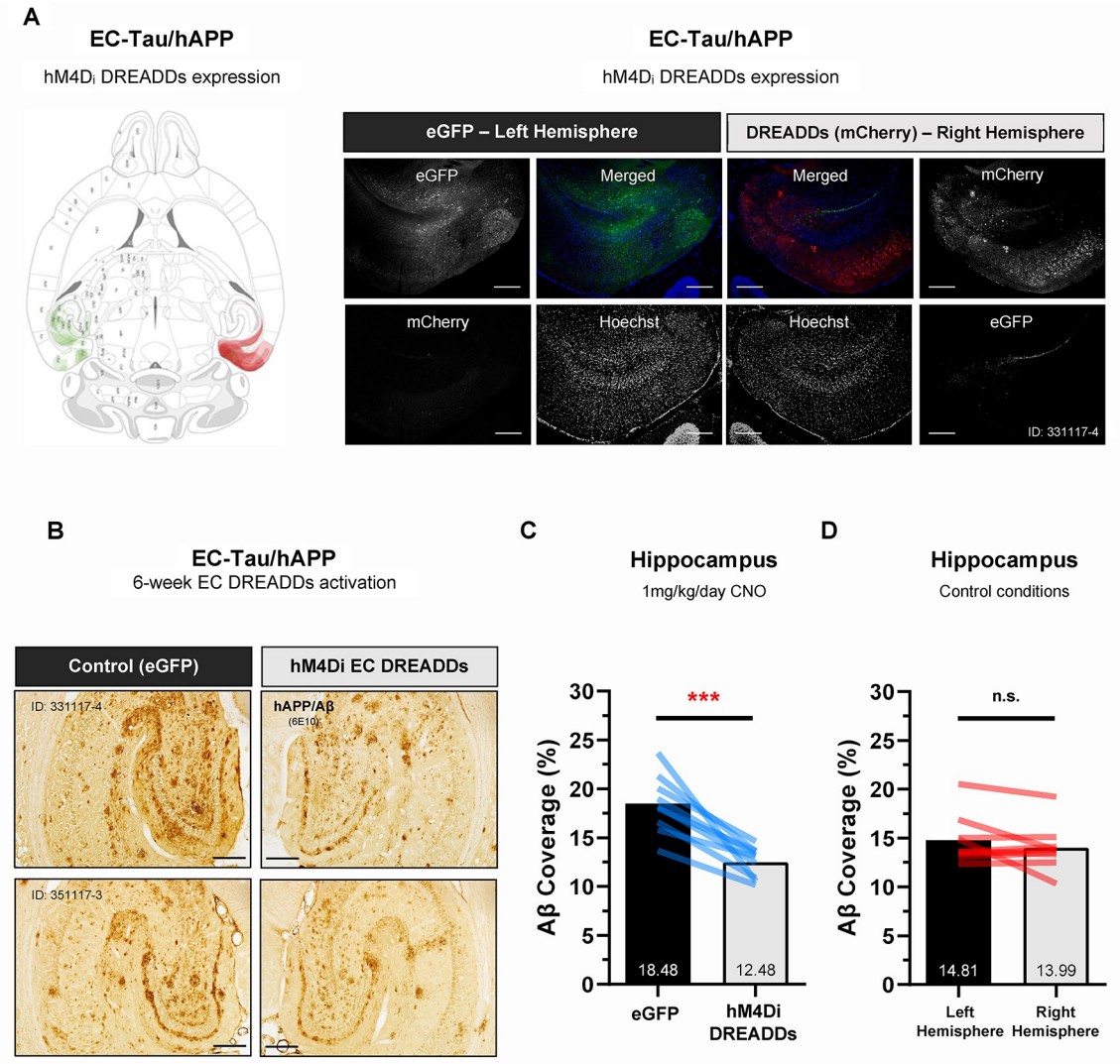

**Fig 5. Chronic hM4D$_i$ EC DREADDs activation reduces Aβ pathology in downstream hippocampus.** Immunohistochemistry was performed on horizontal brain sections from 16-month EC-Tau/hAPP mice subjected to 6 weeks of hM4D$_i$ EC DREADDs activation via osmotic minipump (CNO, 1 mg kg$^{-1}$ day$^{-1}$). **A.** An overlay of individual mCherry (hM4D$_i$ DREADDs) and eGFP (control virus) expression patterns for EC-Tau/hAPP mice ($n$ = 9) is shown. hM4D$_i$ DREADDs expression in the right hemisphere was localized to cell bodies and neuropil throughout the EC, pre- and parasubiculum and subiculum. Representative immunofluorescence images from a 16-month EC-Tau/hAPP mouse are shown. Scale bars, 250 μm. **B-C.** High-magnification images of hippocampal 6E10 + immunostaining are shown for 2 EC-Tau/hAPP mice after chronic hM4D$_i$ EC DREADDs activation. Reduced 6E10 + immunostaining was observed in the right hippocampus versus the contralateral left hippocampus. Semiquantitative analysis of 6E10 + immunoreactivity (Aβ coverage percentage) revealed a significant reduction in hAPP/Aβ in the right-versus-left hippocampus of EC-Tau/hAPP mice ($n$ = 9 total: $n$ = 6 female, $n$ = 3 male). Paired $t$-test: $t$ (8) = 5.712, $p$ < 0.001. Scale bars, 250 μm. **D.** Hippocampal 6E10+ immunoreactivity did not differ between right-versus-left hemispheres of mice in control conditions ($n$ = 9 total: $n$ = 5 female, $n$ = 4 male) (described in Materials and methods). Paired $t$-test: $t$ (8) = 1.357, $p$ > 0.05. All graphical representations appear as mean ± SEM. Colored bar overlays depict the mean percentage values for 6E10+ immunostaining in right-versus-left hippocampal ROIs from 3 brain sections per mouse. ***$p$ < 0.001. Source data are available in S1 Data. Aβ, amyloid-beta; CNO, clozapine-n-oxide; DREADD, designer receptor exclusively activated by a designer drug; EC, entorhinal cortex; eGFP, enhanced green fluorescent protein; hAPP, human amyloid precursor protein; hM4D$_i$, human M4 DREADD receptor; LFP, local field potential; n.s., not significant; ROI, region of interest; SEM, standard error mean.

expressing the P301L mutation resulted in enhanced spread of hTau in proximal and distal brain regions [13]. However, it is unclear whether attenuation of neuronal activity or attenuation of Aβ-associated hyperactivity in the case of EC-Tau/hAPP mice can ameliorate local tau pathology or arrest its spread in vivo. To address these questions, we performed a series of IHC staining experiments on horizontal brain sections from chronic hM4D$_i$ EC DREADDs-activated EC-Tau/hAPP mice ($n = 6$) that exhibited tau pathology within the hippocampus.

We first identified tau aggregates in our tissue by staining for the total human tau marker CP27 (human Tau$^{130-150}$). CP27+ immunoreactivity was clearly present within somatodendritic compartments of neurons along the EC-HIPP network, including the medial and lateral entorhinal cortices, granule cells of the DG and pyramidal CA1 cells in the hippocampus (Fig 6A and 6B). Mice with increased tau pathology exhibited CP27+ immunoreactivity within adjacent perirhinal cortex and within additional subregions of the hippocampus, including the DG hilus and CA3. Semiquantitative threshold analysis of CP27+ immunoreactivity was then performed on high-magnification images of the right and left DG, CA1, and Sub to identify within-subject differences across brain hemispheres (Fig 6C and 6D). A right-versus-left-hemisphere ratio was generated for each region and tissue section analyzed and averaged per mouse. Ratios less than 1.0 indicate reduced CP27+ immunoreactivity in the DREADDs-activated right hemisphere, whereas values greater than 1.0 indicate reduced CP27+ immunoreactivity in the control left hemisphere. A value of 1.0 indicates no hemisphere differences in tau accumulation. We did not detect a significant effect of chronic 1 mg kg$^{-1}$ day$^{-1}$ CNO treatment on total human tau within right-versus-left hippocampal subregions (Fig 6E). Adjacent tissue sections were then processed from our mice to identify phosphorylated tau using the phospho-tau-specific antibody AT8 (Ser$^{202}$/Thr$^{205}$) [52]. Similar to CP27+ immunostaining, AT8 + immunoreactivity was evident within somatodendritic compartments of neurons in the medial and lateral entorhinal cortices and hippocampus (Fig 6F and 6G), with high-magnification images of the DG, CA1, and Sub showing decreased AT8+ staining in the right hippocampus versus left (Fig 6H and 6I). Semiquantitative threshold analysis of AT8+ immunoreactivity in hippocampal ROIs revealed reduced AT8+ staining within the right DG ($p < 0.05$) and right CA1 ($p < 0.05$) but not the right Sub ($p > 0.05$) (Fig 6J). Finally, we processed additional adjacent tissue sections from our mice using the conformation-dependent antibody MC1, which recognizes pathological, abnormally conformed tau [53]. The distribution pattern of MC1+ immunostaining in the brain followed that of CP27 and AT8, with marked MC1+ signal within somatodendritic compartments of neurons along the EC-HIPP network and adjacent cortical and hippocampal subregions (Fig 6K and 6L). However, unlike in AT8+ immunostained brain sections, hemispheric differences in MC1+ immunostaining did not appear as strong under higher magnification (Fig 6M and 6N and S4D Fig). This was confirmed after semiquantitative threshold analysis within hippocampal ROIs, which revealed reduced MC1 + signal in right DG ($p < 0.01$) but not right CA1 ($p > 0.05$) or right Sub ($p > 0.05$) (Fig 6O). These data suggest that chronic attenuation of neuronal activity in upstream neurons can have an impact on the accumulation of phospho-tau and abnormally conformed tau in synaptically connected, downstream neurons but that reducing the propagation of total human tau may be more difficult.

In conclusion, our experimental data strongly support the emerging hypothesis that hAPP/ Aβ accumulation in the brain is associated with aberrant neuronal activity and network impairment. Our data describe a role for Aβ-associated neuronal hyperactivity in accelerating tau pathology along a well-characterized neuronal network that is vulnerable to AD pathology and neurodegeneration. We further show that hyperactivity in this network can be targeted via chronic chemogenetic activation to arrest the accumulation of both hAPP/Aβ and pathological tau along the EC-HIPP network in vivo.

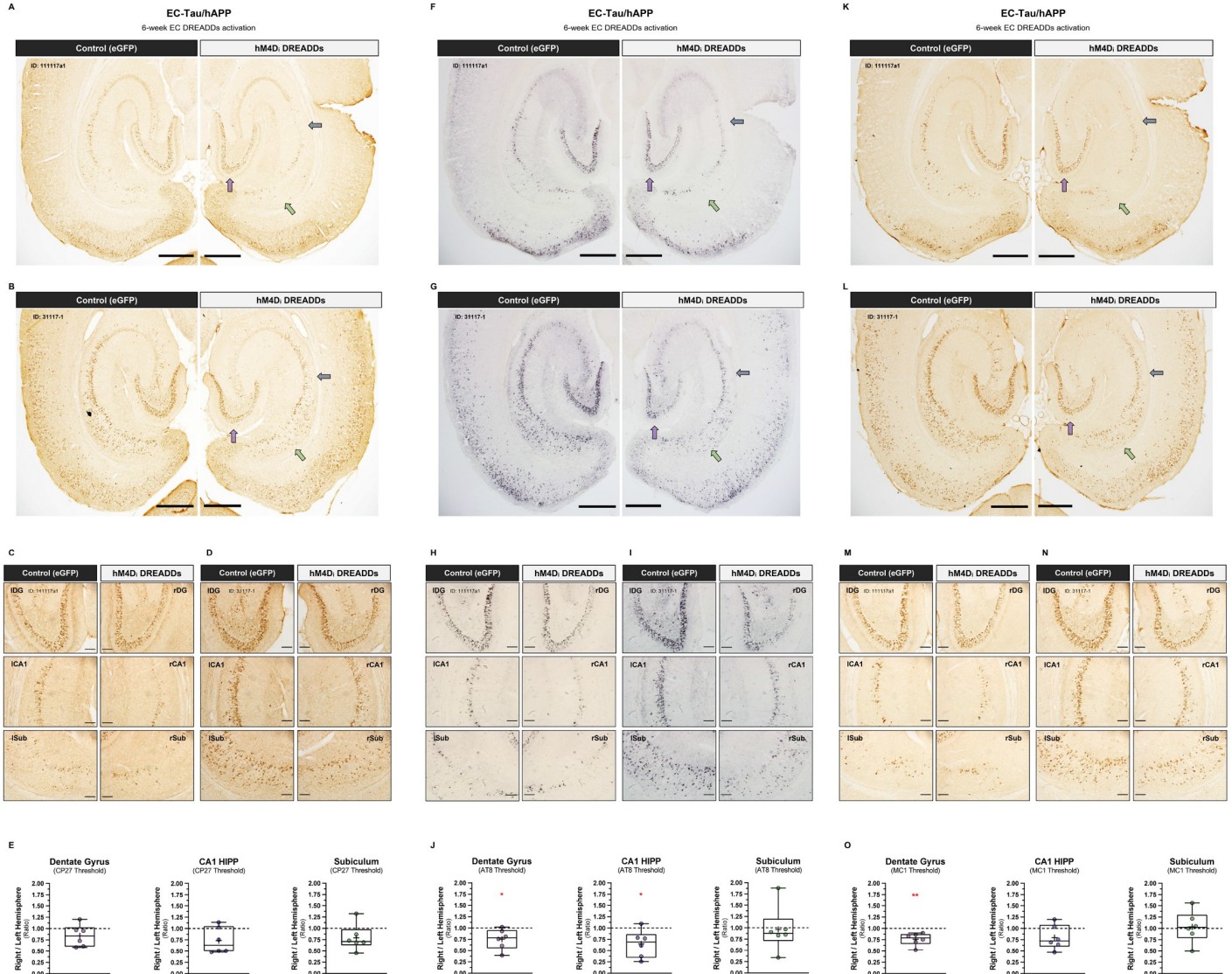

**Fig 6. Chronic hM4D$_i$ EC DREADDs activation reduces tau pathology in downstream hippocampal subregions.** Semiquantitative analysis of tau pathology expression in the hippocampus was performed on CP27+, AT8+, and MC1+ immunostained brain sections from 16–18-month EC-Tau/hAPP mice after chronic hM4D$_i$ EC DREADDs activation (total, $n = 6$: female, $n = 2$; male, $n = 4$). **A-B.** Low-magnification images of CP27+ immunostained brain sections from 2 EC-Tau/hAPP mice are shown. Colored arrows denote the 3 hippocampal subregions wherein total human tau was examined: DG (purple), CA1 (blue), and Sub (green). Mouse IDs appear at top left. Scale bars, 500 μm. **C-D.** Higher-magnification images of the right-versus-left DG, CA1, and Sub are shown. Scale bars, 100 μm. **E.** Tau immunoreactivity was calculated per hippocampal ROI in both the right and left hemisphere of each tissue section, and then plotted as a ratio of right over left hemisphere values to reveal laterality in pathological tau accumulation. No significant differences in total human tau staining (CP27+) were detected in hippocampal subregions across hemispheres. One-sample $t$-tests: (DG, CP27 threshold) $t_{(5)} = 0.1612$, $p = 0.1678$; (CA1, CP27 threshold) $t_{(5)} = 2.253$, $p = 0.0740$; (Sub, CP27 threshold) $t_{(5)} = 1.740$, $p = 0.1423$. **F-G.** Low-magnification images of AT8+ immunostained brain sections from 2 EC-Tau/hAPP mice are shown (same as in panels A-D). Scale bars, 500 μm. **H-I.** Higher-magnification images of the right-versus-left DG, CA1, and Sub are shown. Scale bars, 100 μm. **J.** AT8+ immunostaining for hyperphosphorylated tau revealed decreased tau accumulation in the DG and CA1 subregions in the right hemisphere, downstream from hM4D$_i$ EC DREADDs expression. One-sample $t$-tests: (DG, AT8 threshold) $t_{(5)} = 2.632$, $p = 0.0464$; (CA1, AT8 threshold) $t_{(5)} = 2.807$, $p = 0.0377$; (Sub, AT8 threshold) $t_{(5)} = 0.175$, $p = 0.8679$. **K-L.** Low-magnification images of MC1+ immunostained brain sections from the same 2 EC-Tau/hAPP mice are shown. Scale bars, 500 μm. **M-N.** Higher-magnification images of the right-versus-left DG, CA1, and Sub are shown. Scale bars, 100 μm. **O.** MC1+ immunostaining for abnormally conformed tau revealed decreased tau accumulation in the DG of the right hemisphere, downstream from hM4Di EC DREADDs expression. One-sample $t$-tests: (DG, MC1 threshold) $t_{(5)} = 4.213$, $p = 0.0084$; (CA1, MC1 threshold) $t_{(5)} = 1.827$, $p = 0.1272$; (Sub, MC1 threshold) $t_{(5)} = 0.238$, $p = 0.8212$. Individual mouse right/left brain hemisphere ratios appear as colored overlays and represent averaged values from 2 immunostained sections per mouse per tau antibody. Mean values appear as a "+" within boxplots. $^*p < 0.05$, $^{**}p < 0.01$. Source data are available in S1 Data. Aβ, amyloid-beta; AT8, Ser$^{202}$/Thr$^{205}$ phosphotau-specific antibody; CA1, Cornu Ammonis 1; CNO, clozapine-n-oxide; CP27+, human Tau$^{130-150}$ antibody; DG, dentate gyrus; DREADD, designer receptor exclusively activated by a designer drug; EC, entorhinal cortex; eGFP, enhanced green fluorescent protein; hAPP, human amyloid precursor protein; HIPP, hippocampus; hM4D$_i$, human M4 DREADD receptor; ID, identification; LFP, local field potential; MC1, antibody for misfolded tau; n.s., not significant; ROI, region of interest; Sub, subiculum.

## Discussion

The progressive, stereotypical spread of pathological tau along neuronal circuits in AD is an active area of intense investigation. We (and others) have proposed that increased neuronal activity can exacerbate tauopathy either by promoting tau release from neurons or by facilitating its uptake in synaptically connected neurons [32, 37]. To model the functional interactions between hAPP/Aβ and hTau in a well-characterized neuronal circuit in vivo, we crossed the Aβ-generating hAPP/J20 (Swedish/Indiana) mouse line [21, 54] with the EC-Tau mouse line, wherein mutant hTau$^{P301L}$ is predominantly expressed in the EC [10, 11]. Network dysfunction and impaired cognitive performance have previously been described in relatively young hAPP mice [21, 22, 48]. Mutant hAPP expression in the EC-Tau/hAPP mouse thus provides a strong in vivo platform to model the impact of Aβ-associated hyperactivity on tau pathology. At 16 months, somatodendritic MC1+ immunostaining was increased in the EC and DG (Fig 1G and 1H). By 23 months, the distribution and severity of MC1+ immunostaining resembled that of much older (30+ month) EC-Tau mice (Fig 1B) [39, 55], suggesting that hAPP/Aβ plays a significant role in promoting the intercellular transfer of pathological tau in vivo. Our data supports previously published findings in similar mouse models, in which robust hAPP/Aβ accumulation was associated with accelerated tauopathy along the EC-HIPP network [36–38].

We hypothesized that increased Aβ production or accumulation may trigger an intermediate, nonpathogenic cascade of events that impact tau. Indeed, substantial evidence exists to support the hypothesis that increased Aβ leads to neuronal hyperactivity and large-scale network dysfunction in the brain [24]. This hypothesis may partially explain the accelerated progression of pathological tau in AD and in mouse models of AD pathology. hAPP/Aβ accumulation has been linked to the appearance of epileptiform-like network activity in the brain at a relatively young age [21, 22, 56]. This aberrant network activity is associated with changes in inhibitory neuron profiles and remodeling of the DG. More recently, hyperactive neurons have been shown to disproportionately cluster around Aβ plaques in cortex [20], in which elevated concentrations of soluble Aβ oligomers may collect [28]. Motivated by these observations and our findings (Fig 1), we examined single-unit activity and LFPs in the EC of 16-month EC-Tau/hAPP mice and their age-matched littermates (Figs 2 and 3). EC-Tau/ hAPP mice and hAPP mice exhibited significant EC single-unit hyperactivity and network dysfunction, characterized by increased average firing rates (Fig 2B), decreased median ISIs (Fig 2C), and decreased percentage theta power compared with control mice (Fig 3A–3C). Consistent with a greater susceptibility for seizure-like activity in hAPP mice, we found increased neuronal bursting activity in hAPP mice versus control mice (S1 Data, 1-way ANOVA: $F_{(3)}$ = 4.887, $p < 0.01$. Dunnett's multiple comparisons) [57]. Plotting the average firing rates of pooled EC neurons as a function of spike width revealed a distinct population of putative interneurons (NS) and excitatory cells (WS) in our dataset (Fig 2D). EC-Tau/hAPP mice and hAPP mice each exhibited hyperactive WS neurons, whereas only hAPP mice showed hyperactive NS cells (Fig 2E and 3F). Aβ-associated EC hyperactivity and theta impairment was also present after applying a speed filter to the datasets, which removed neuronal activity during bouts of immobility (S1 Fig). These data support previous findings describing Aβ-associated interneuron dysfunction and network hyperactivity [21, 22, 58] and extend them to putative inhibitory and excitatory neurons in EC. hAPP/Aβ-associated disruptions in theta oscillatory activity are also in line with previous reports [43, 45]. Finally, we did not detect hAPP/Aβ-mediated effects on behavioral activity during open field recording sessions (S2 Fig). This is in contrast to several reports describing locomotor hyperactivity in the hAPP/J20 mouse line [22, 48, 49]. We predict that the discrepancy may be due to increased

motivational drive to actively explore the arena in our mice, as sucrose pellets were administered during recording sessions to encourage foraging behavior and arena coverage. Repeated handling and acclimation to the recording paradigm may have also reduced innate anxiety-like behavior in our mice, leading to similar behavioral activity patterns. Importantly, we can conclude that Aβ-associated neuronal hyperactivity and network dysfunction are not due to gross locomotor differences in our mice.

Several lines of evidence now support a role for increased neuronal activity in both Aβ accumulation and accelerated tau pathology in vivo. Stimulation of the perforant pathway results in increased Aβ concentrations in hippocampal interstitial fluid (ISF) [30] and increased Aβ deposition in downstream DG (outer molecular layer) [33]. Chemogenetic stimulation of cortical neurons is also associated with increased deposition of mature amyloid plaques [34]. Likewise, stimulating neuronal activity results in increased ISF hTau concentrations [31], and chronic stimulation of EC neurons enhances local tauopathy and accelerates neurodegeneration [32]. These data suggest that Aβ-associated hyperactivity can impact pathological tau progression along a defined neural circuit while simultaneously driving increased hAPP/Aβ release. The afflicted circuit could then potentially be recruited into a harmful, positive feedback loop that drives aggressive pathological hAPP/Aβ and tau aggregation in the neuronal network, leading to cognitive impairment and cell death. Aged mice with significant tauopathy along the EC-HIPP network exhibit deficits in spatially modulated grid cell function and impaired spatial learning and memory, as well as excitatory neuron loss, independent of Aβ pathology [39, 55].

Surprisingly, the increased aggregation of pathological tau in EC did not appear to affect neuronal activity measures of hyperactivity or network function in 16-month EC-Tau mice. Single-unit average firing rates, ISI medians, and percentage theta power were remarkably similar in EC-Tau/hAPP versus hAPP mice and in EC-Tau versus nontransgenic control mice. These findings support a previous report from our lab describing normal EC single-unit function in 14-month EC-Tau mice, in which average firing rates and spatially modulated cell firing patterns matched that of nontransgenic controls [39]. These data suggest that the regional accumulation of tau in this particular mouse model does not strongly impact EC neuronal activity by 16 months in vivo. However, these data should be carefully interpreted, as mutant hTau$^{P301L}$ overexpression in EC has been linked to hypometabolism in approximately 9-month-old mice [36] and has been shown to blunt Aβ-associated hyperexcitability in vitro [59]. When mutant hTau$^{P301L}$ is overexpressed throughout excitatory forebrain neurons (rTg4510 mouse line), dysfunctional neuronal oscillatory patterns are observed along with reduced neuronal firing rates [35]. More recently, hTau$^{P301L}$ overexpression in relatively young rTg4510 mice was associated with significant neuronal silencing in cortex and suppression of Aβ-associated hyperactivity using in vivo Ca$^{2+}$ imaging [60]. Increased hTau$^{P301L}$ hyperphosphorylation may be responsible for reduced neuronal activity in these mice, as relatively young rTg4510 mice exhibit impaired neuronal excitability linked to structural changes to the axon initial segment (AIS) in a microtubule-dependent manner [61]. Interestingly, the frontotemporal dementia (FTD)-associated hTau$^{V337M}$ mutation was recently shown to increase neuronal excitability in human induced pluripotent stem cell–derived neurons by impacting AIS activity-dependent plasticity [62]. These data agree with several reports showing human mutant tau variants are associated with network hyperexcitability in mouse models [63–67]. It is possible then that detection of hTau-mediated effects on neuronal activity is partly dependent on the sensitivity of the assays used to measure them, the type of tau being studied (e.g., P301L, V337M, etc) and the degree of pathological severity in the chosen model system. Our analysis in 16-month EC-Tau/hAPP mice revealed blunted hyperactivity in NS neurons (Fig 2E). This may represent a relatively early effect of hTau$^{P301L}$ on Aβ-associated

interneuron dysfunction prior to subsequent impairment in excitatory neurons. For instance, we have shown that aged (30+ month) EC-Tau mice exhibit significant hypoactivity in excitatory MEC grid cells [39]. Thus, we predict that divergent hAPP/Aβ and hTau effects on neuronal activity would be readily observed in the EC-HIPP network of EC-Tau/hAPP mice aged beyond 16 months.

We implemented a chemogenetic approach in our studies to combat the aggressive progression of Aβ-associated EC hyperactivity on Aβ and tau pathology in vivo. hM4D$_i$ DREADDs were targeted to EC principal neurons based on the finding that WS neurons showed hyperactivity in both hAPP mice and EC-Tau/hAPP mice (Fig 2F). DREADD-mediated neuromodulation has previously been shown to reduce local Aβ deposition in cortex (hM4D$_i$) [34] and facilitate the transfer of hTau into distal postsynaptic cell populations using excitatory DREADD receptors(hM3D$_q$) [13]. Using a within-subjects experimental design, we probed for hAPP/Aβ and pathological tau immunoreactivity in both the ipsilateral hippocampus (downstream from hM4D$_i$ EC DREADDs expression) and in the contralateral hippocampus. After 6 weeks of hM4D$_i$ EC DREADDs activation, we found a marked reduction in 6E10 + immunoreactivity within ipsilateral hippocampus of EC-Tau/hAPP mice (Fig 5B and 5C), supporting previous reports linking stimulated neuronal activity to Aβ release and Aβ pathology [30, 33, 34, 68, 69]. Chronic hM4D$_i$ EC DREADDs activation in EC-Tau/hAPP mice also led to selective reductions in pathological tau immunoreactivity within ipsilateral, right hemisphere DG, and CA1 hippocampal subfields versus the contralateral left hemisphere (Fig 6). Specifically, we found significant reductions in hyperphosphorylated tau in DG and CA1 using the AT8+ antibody (Fig 6F–6J) and a significant reduction in abnormally conformed tau using MC1+ (Fig 6K–6O). One-sample $t$-tests were run individually for each marker and hippocampal subfield to compare right-versus-left ratio group means to a hypothetical mean of 1.0, which would indicate equal distribution of pathological tau in right and left hippocampal ROIs. Our results suggest that chronic chemogenetic attenuation of neuronal activity can reduce the accumulation of pathological tau in vivo and supports previous reports showing that stimulated neuronal activity can increase tau release and tauopathy in AD mouse models [12, 13, 31, 32]. Importantly, chronic hM4D$_i$ EC DREADDs activation reduced, but did not eliminate, hallmark AD pathologies in 16-month EC-Tau/hAPP mice (Figs 5 and 6). It is possible then that a longer regimen of DREADDs activation and earlier intervention, prior to the onset of Aβ plaques and tau aggregation, would increase the preservation of EC-HIPP circuit function in the EC-Tau/hAPP mouse model.

Additional studies will be required to determine the precise mechanisms mediating pathological Aβ and tau reduction after chronic chemogenetic activation. However, we speculate that several mechanisms may underlie the effects observed in our study. First, we hypothesize that chronic attenuation of EC neuronal activity suppresses both activity-dependent Aβ and tau production/secretion from EC neurons, slowing the progressive accumulation of these pathologies along the ipsilateral EC-HIPP circuit [12, 13, 30–34, 70]. This may in turn provide a temporal window of opportunity for increased clearance of soluble, prefibrillar forms of Aβ and tau in the ipsilateral hippocampus. Our Figs 5 and 6 results may then be a product of (1) decreased Aβ and tau production/release and (2) increased Aβ and tau clearance via several potential mechanisms. It is possible that microglial reactivity is enhanced in our mice after DREADDs activation, either by increasing proteolytic clearance of Aβ surrounding plaques or by promoting a barrier function to prevent the outward extension of amyloid fibrils [71]. This may occur despite no reported effect on the overall distribution or gross morphology of microglial cells after DREADDs activation using the Iba1 antibody [34]. Importantly, microglial reactivity has also been observed to exacerbate tau pathology and contribute to tau spread in mouse models [72, 73]. These findings underscore the importance of additional experiments

examining the role of microglia in mouse models of Aβ and tau pathology. Chronic chemogenetic activation may have also facilitated glymphatic-mediated clearance of Aβ and tau in our mouse model, though this has not been investigated. The glymphatic system is a highly organized fluid transport system responsible in part for metabolic waste removal from the brain parenchyma and has been implicated in clearance of Aβ and tau during sleep [74–76]. Thus, investigation of sleep–wake cycles and glymphatic activity in EC-Tau/hAPP mice during chronic DREADDs activation is warranted. Finally, attenuation of EC neuronal activity using DREADDs may have also improved the activity of classical protein degradation pathways (e.g., proteasome, lysosome) in EC-Tau/hAPP mice.

Recent concerns have been raised in the literature regarding the utility of CNO as an inert DREADDs ligand that easily permeates the blood–brain barrier (BBB) [77–79]: for review, see [80]. In our studies, we first confirmed chronic EC DREADDs activation in vivo using recording metrics derived from single, high-dose CNO injection studies. Acute hM4D$_i$ DREADDs activation resulted in decreased EC spiking activity, with a maximal response at approximately 40–60 minutes that lasted for at least 4 hours (S3A and S3B Fig). These data are supported by previous in vivo research showing strong hM4D$_i$ activation in EC 30 minutes post-CNO, with activity levels recovering toward baseline by 12 hours [81]. Acute hM4D$_i$ DREADDs activation also led to a decrease in percentage theta power in EC (S3C and S3D Fig). Therefore, we tracked chronic EC DREADDs activation in vivo by analyzing percentage theta power and total spike counts (Fig 4A and 4B), rather than measuring levels of CNO or converted clozapine in peripheral blood/plasma. Consistent with our single CNO injection findings, percentage theta power and total spike counts were reduced over chronic CNO treatment relative to baseline measures. Further analysis revealed that average firing rates were significantly reduced in EC neurons by week 6 of chronic CNO treatment (Fig 4C) and that despite targeting hM4D$_i$ DREADDs to CamKII$_\alpha$-expressing excitatory neurons, both NS and WS neurons were impacted (Fig 4D and 4E). This may be expected given that EC excitatory and interneurons are disynaptically connected [82]. Collectively, these data support the utility of long-term CNO delivery in indwelling osmotic minipumps to activate DREADDs in vivo (see also [83]). Importantly, percentage theta power was further reduced in EC-Tau/hAPP mice following chronic hM4D$_i$ DREADDs activation. These mice already exhibit decreased theta power compared with control mice (Fig 3C), suggesting that additional experiments will be necessary to find an optimal degree of neuronal circuit manipulation to restore network activity to nontransgenic control levels.

Expression of hM4D$_i$ EC DREADDs was restricted to the right hemisphere of our mice and did not crossover into the contralateral, left hemisphere (Fig 5A). Thus, we were able to discriminate the effects of chronic EC DREADDs activation on pathology in ipsilateral, downstream hippocampus (right hemisphere) and directly compare it to pathology in the contralateral, control left hippocampus. We predict that any off-target effects of chronic, converted CNO-to-clozapine on hAPP/Aβ and tau pathology would have impacted both left and right hemispheres in our experimental mice, as continuous 6-week delivery of 1 mg kg$^{-1}$ day$^{-1}$ CNO was performed using minipumps. hAPP/Aβ (6E10+) immunostaining revealed strong decreases in hAPP/Aβ in the ipsilateral right hippocampus (Fig 5B and 5C), downstream of the DREADDs-activated right EC. We did not detect hemisphere differences in 6E10+ immunoreactivity in age-matched hAPP mice or EC-Tau/hAPP mice sampled from our colony (no DREADDs manipulation) or in hM4D$_i$ EC DREADDs expressing EC-Tau/hAPP mice administered a lower CNO dose (0.5 mg kg$^{-1}$ day$^{-1}$) (Fig 5D). Furthermore, decreases in phosphotau and abnormally conformed tau were detected in ipsilateral, right hippocampal subfields after chronic hM4D$_i$ EC DREADDs activation (Fig 6). The degree to which pathological tau and Aβ accumulation were reduced in downstream HIPP,

along with our electrophysiology data showing reduced neuronal activity, is consistent with the hypothesis that approximately 50%–75% maximal DREADDs activation could be achieved with 1 mg kg$^{-1}$ day$^{-1}$ CNO [84].

Attenuating neuronal hyperactivity and network dysfunction may prove to be a powerful tool in combating impaired cognition in human AD, especially when paired with therapies aimed at alleviating the aggregation and deposition of Aβ and tau. On its own, Aβ-targeted immunotherapy has proven unsuccessful at relieving AD cognitive symptoms, which may be due to ineffective amelioration or the exacerbation of neuronal hyperactivity [29, 85]. Indeed, reducing AD pathology-associated neuronal dysfunction with the antiepileptic drug leviterace-tam has shown promise in preclinical mouse models of hAPP overexpression [22, 86], and is being tested for efficacy in AD clinical trials. Our data show that alleviating Aβ-associated EC hyperactivity in a transgenic mouse model reduces downstream accumulation of both Aβ and tau pathology in the hippocampus. An important step forward will be to replicate these findings in nonoverexpressing hAPP mice ($App^{NL-F/NL-F}$), which show early signs of neuronal hyperexcitability in vitro and impaired gamma oscillations in vivo [87–89]. In addition, future studies will be required to determine if relief from pathological Aβ and tau using approaches such as chemogenetics will improve cognitive behavior once aberrant neuronal activity is returned to wild-type control levels. Interestingly, depletion of murine tau has been shown to alleviate locomotor and neuronal network hyperactivity in young hAPP/J20 mice and can blunt chemically induced aberrant overexcitation [49, 90–92]. In APP/PS1+Tau mice (rTg21221), suppressing human tau transgene expression also alleviates locomotor hyperactivity in these mice in addition to ameliorating key gene expression changes involved in neuroinflammation and synaptic function [93]. This would indicate that several potential mechanisms exist in the brain to contribute to impaired neuronal activity in AD and provide ample avenues for investigation into the etiology of AD progression.

## Materials and methods

### Experimental animals and ethics statement

Three transgenic mouse lines were used in these studies to model hallmark AD pathologies in the brain. First line: bigenic EC-Tau mice that overexpress human mutant tau (4R0N P301L) predominantly in the EC (via neuropsin promoter) on a C57BL/6J and FVB/N background, described in detail previously [10, 11, 94]. Second line: hAPP/J20 mice that overexpress hAPP with 2 familial AD mutations (KM670/671NL, Swedish) (V717F, Indiana) on a C57BL/6J background [54, 95]. Third line: EC-Tau/hAPP mice, which generate both hAPP/Aβ and EC-specific tau pathology in the brain. This triple transgenic mouse line was created by crossing bigenic EC-Tau mice with hAPP/J20 mice. Because of increased mortality rates in pups born to hAPP/J20 carrier females, hAPP/J20 males were bred with EC-Tau females to generate all mice used in these studies. All mouse lines are heterozygous.

A total of 58 mice, including males ($n = 27$) and females ($n = 31$), were used as experimental animals in these studies. Total mouse numbers per genotype were as follows: nontransgenic controls (control, $n = 10$), and age-matched, transgenic littermates (EC-Tau, $n = 17$; hAPP, $n = 12$; EC-Tau/hAPP, $n = 19$). All mice were housed in a temperature and humidity-controlled vivarium at Columbia University Medical Center and maintained on a standard 12-hour light/dark cycle with food and water provided ad libitum. All animal experiments were performed during the light phase in accordance with national guidelines (National Institutes of Health) and approved by the Institutional Animal Care and Use Committee of Columbia University (IACUC Approval Number: AAAS1500).

## Microdrive construction

Microdrives were constructed as described previously [39, 96]. Briefly, custom-made reusable 16-channel or 32-channel microdrives (Axona, UK) were outfitted with 4 to 8 tetrodes consisting of twisted, 25-μm-thick platinum-iridium wires (California Wires, USA) funneled through a 23-gauge stainless steel inner cannula. A 19-gauge protective, stainless steel outer cannula was slipped over the inner cannula and secured to the microdrive body via modeling clay. Individual electrodes were wrapped tightly around the exposed wires of the microdrive and coated with a layer of Pelco conductive silver paint (Ted Pella, Inc., USA) prior to sealing of the microdrive body with liquid electrical tape (Gardner Bender, USA). Several hours prior to surgery, the tetrodes were cut to an appropriate length and electroplated with a platinum/gold solution until the impedances dropped within a range of 150–200 Kohms.

## DREADDs virus microinjection and tetrode implantation

A total of 31 mice were implanted and recorded for single-unit in vivo electrophysiology studies shown in Fig 2 and S1 Fig (control, $n = 8$; EC-Tau, $n = 7$; hAPP, $n = 8$; EC-Tau/hAPP, $n = 8$). LFP data in Fig 3 and S1 Fig were recorded from a total of 37 mice (control, $n = 9$; EC-Tau, $n = 8$; hAPP, $n = 10$; EC-Tau/hAPP, $n = 10$).

On the day of electrode implantation, mice were anesthetized with isoflurane (3%–4% for induction; 0.5%–3% for maintenance) using a multichannel VetFlo Traditional Anesthesia vaporizer (Kent Scientific) and fixed within a stereotaxic frame (Kopf Instruments). As described previously [39], an incision was made to expose the skull, and 3 jeweler's screws were inserted into the skull to support the microdrive implant. A 2-mm hole was made on each side of the skull at position 3.0–3.1 mm lateral to lambda and approximately 0.2 mm in front of the transverse sinus. Viral delivery of the $G_i$-coupled $hM4D_i$ DREADDs (AAV5-CamKII$_\alpha$-hM4D$_i$-mCherry, $4.1\times10^{12}$) (Cat No. 50477-AAV5; Addgene, USA) was administered into the right EC via a 33-gauge Neuros Syringe (Hamilton, USA) tilted at an angle of 6˚–7˚ in the sagittal plane. A control virus (AAV5-CamKII$_\alpha$-eGFP, $4.1\times10^{12}$) (Cat No. 50469-AAV5; Addgene, USA) was administered into the contralateral left EC using identical methods and measurements. At this point, an additional screw connected with wire was inserted into the skull, serving as a ground/reference for LFP recordings. The prepared microdrive was then tilted at 6˚–7˚ on a stereotaxic arm and the tetrodes lowered to 1.0 mm from the surface of the brain (below dura) into the DREADDs-delivered right hemisphere. The microdrive ground wire was then soldered to the skull screw wire and the microdrive was secured with dental cement. Mice were then allowed to recover in a cleaned cage atop a warm heating pad until awake (approximately 45 minutes) before being transported to the housing facility. Mice received Carprofen (5 mg/kg) prior to surgery and postoperatively to reduce pain, in additional to a sterile saline injection (s.c.) to aid in hydration. Recording experiments began approximately 1 week from the time of surgery.

## In vivo recording: single-unit and LFP analyses

For recording experiments, mice from each genotype were run in parallel and balanced where possible. All mice outfitted with microdrives underwent 4 to 6 recording sessions in an arena (70 cm × 70 cm), with 1 recording session performed per day. Tetrodes for each mouse were moved down 100 μm from their previous position 24 hours prior to the next recording session, allowing stable electrode positioning and a robust sampling of EC neuronal activity for each mouse. Additionally, the arena and visual cue were rotated between sessions.

Neuronal signals from our mice were recorded using the Axona DacqUSB system and described previously [39]. Briefly, recording signals were amplified 10,000 to 30,000 times and

bandpass filtered between 0.8 and 6.7 kHz. The LFP was recorded from 4 channels of each microdrive, amplified 8,000 to 12,000 times, lowpass filtered at 125 Hz, and sampled at 250 Hz. Electrical noise (60-Hz) was eliminated using a Notch filter. A minimum–maximum speed filter (5–30 cm/seconds) was applied to LFP data where noted. Spike sorting was performed offline using TINT cluster-cutting software and KlustaKwik automated clustering tool. The resulting clusters were further refined manually and were validated using autocorrelation and cross-correlation functions as additional separation tools. Only cells that produced a minimum of 100 spikes with refractory periods greater than 1 millisecond were used for subsequent analysis [41]. Single units with no undershoot in their waveform were also discarded. Putative excitatory neurons (WS) were distinguished from putative interneurons (NS) by first examining the frequency distribution histogram of pooled EC single-unit spike widths and then bisecting the waveform spike widths after the first modal distribution. This resulted in a delineation cutoff at 340 μs (Fig 2D) [39, 41]. Quantitative measurements of cluster quality were then performed, yielding isolation distance values in Mahalanobis space [97]. Median isolation distance values were then calculated per mouse followed by comparison across genotypes. There were no significant differences detected between clusters in our experimental groups (isolation distances: control, 7.795; EC-Tau, 8.270; hAPP, 8.900; EC-Tau/hAPP, 7.805: 1-way ANOVA, $F = 2.531$, $p > 0.05$). Finally, average firing rates were calculated by dividing the total number of spikes by duration of recording session, with a minimum–maximum speed filter (5–100 cm/seconds) applied where noted. A burst was defined as consisting of no less than 2 spikes with a maximum ISI of 10 milliseconds. The percentage of spike bursts of every cell in a session was calculated per mouse, and average values were compared across genotypes.

A total of 1,910 EC neurons were recorded across 31 mice for data shown in Fig 2 and S1 Fig (control, $n = 386$ single units [$n = 8$ mice]; EC-Tau, $n = 404$ [$n = 7$ mice]; hAPP, $n = 532$ [$n = 8$ mice]; EC-Tau/hAPP, $n = 588$ [$n = 8$ mice]). A total of 285 EC neurons were recorded across 8 EC-Tau/hAPP mice for data shown in Fig 4C–4E (baseline: $n = 154$; week 6, CNO: $n = 131$). A complete breakdown of the analyzed data can be found in S1 Data.

## Osmotic minipump preparation and surgical implantation

To chronically activate hM4D$_i$ EC DREADDs receptors in vivo, EC-Tau/hAPP mice ($n = 9$) were implanted with osmotic minipumps and monitored throughout a 6-week treatment regimen. CNO (Sigma Aldrich, USA) was dissolved in sterile saline with 0.05% DMSO. Twenty four hours prior to surgery, osmotic minipumps (Model 2006) (Alzet, USA) were primed with CNO at a concentration range of 7–10 μg/μL in a sterile fume hood (Alzet drug concentration calculator: flow rate, 0.15 μL/hour; target dose, 1 mg kg$^{-1}$ day$^{-1}$). CNO-filled pumps were stored in warm (37°C) sterile saline in a conical tube until surgery.

Mice were anesthetized with isoflurane as previously described, and the fur clipped at the abdomen. A single injection of marcaine (2 mg/kg, 0.05 mL) was delivered intradermally into the site of incision approximately 5 minutes prior to surgery, then a midline incision was made in the abdominal wall and a sterile, CNO-filled minipump was maneuvered into the intraperitoneal (i.p.) cavity. The incision was then closed using an absorbable suture (Henry Schein, USA) for the abdominal layer, followed by closure of the skin with a nylon synthetic nonabsorbable suture (Henry Schein, USA). A topical antibiotic was then administered at the surgical site and the mice were allowed to recover in a cleaned cage atop a warm heating pad until awake (approximately 15 minutes). Mice implanted with minipumps were administered Carprofen (5 mg/kg) prior to surgery and postoperatively to help reduce pain. Nonabsorbable sutures were removed approximately 10 days after surgery.

## DAB immunohistochemistry and immunofluorescence imaging

Six weeks after osmotic minipump implantation, all mice were deeply anesthetized with a cocktail of ketamine/xylazine before being transcardially perfused with ice-cold 100 mM phosphate-buffered saline (PBS, pH 7.4), followed by 10% formalin (Fisher Scientific, USA). The last recording position for each microdrive-implanted mouse was recorded, and then the microdrive removed. Brains were then harvested and left in 10% formalin overnight, then incubated in 30% sucrose until the brains sank to the bottom of the conical tube (all at 4˚C). Horizontal brain sections were sliced (30 μm) using a Leica CM3050 S cryostat and stored in cryoprotectant at -20˚C until immunostaining procedures. EC-Tau/hAPP mice and EC-Tau mice that appear in Fig 1 were euthanized immediately for immunostaining. For both immunoperoxidase staining and immunofluorescence imaging, EC-Tau/hAPP brain sections were processed in parallel with sections from EC-Tau, hAPP, and nontransgenic control mice where appropriate. Finally, all tissue sections included for semiquantitative analysis of hAPP/ Aβ (Fig 5) and tau (Fig 6) were verified to be within the range of DREADDs mCherry expression by first checking for native fluorescence signal in free-floating sections on an inverted Olympus epifluorescence microscope.

Immunoperoxidase staining was performed using a Mouse-on-Mouse kit (Vector Laboratories, USA) and modified from previous methodology [11, 39, 55]. Briefly, cryoprotectant was washed from the free-floating tissue sections with PBS before quenching endogenous peroxidases with 3% $H_2O_2$. Sections were then blocked with mouse IgG–blocking reagent for 1.5 hours at room temperature, followed by overnight incubation at 4˚C with either anti-beta amyloid 6E10 (mouse, 1:1,000 dilution of 1 mg/mL stock) (Biolegend, USA), anti-tau MC1 (mouse, 1:500; courtesy of Peter Davies, Feinstein Institute for Medical Research, North Shore LIJ), biotinylated anti-phospho tau$^{Ser202/Thr205}$ AT8 (mouse, 1:500) (Thermo Scientific, USA) or anti-human tau CP27 (mouse, 1:500; courtesy of Peter Davies). Tissue sections were then rinsed in PBS and incubated for approximately 20 minutes at room temperature in a working solution of biotinylated anti-mouse IgG reagent (excluding the biotinylated AT8-labeled sections). After several PBS rinses, sections were then incubated in an avidin-biotin conjugate for 10 minutes before being developed in $H_2O$ containing 3,3′-diaminobenzidine (DAB) hydrochloride and urea hydrogen peroxide (Sigma Aldrich, USA). After staining was completed, tissue sections were mounted onto glass Superfrost Plus slides (Fisher Scientific, USA), allowed to air dry completely, and then dehydrated in ethanol and cleared with xylenes before being coverslipped.

Horizontal tissue sections used to visualize native DREADDs mCherry and eGFP expression in the brain were first rinsed in PBS containing 0.3% Triton X-100 (Sigma Aldrich, USA) (PBST) and then incubated in a working solution of Hoechst 33342 dye (5 μg/mL) (Thermo Scientific, USA) to stain cell nuclei for 10 min at room temperature. Subsequent washes in PBST were followed by mounting the tissue onto Superfrost Plus slides and coverslipping using SlowFade gold anti-fade reagent (Life Technologies, USA). All slides were stored in the dark at 4˚C until imaging.

## DAB IHC image analysis

Immunoperoxidase-stained tissue sections were analyzed under bright-field microscopy using an Olympus BX53 upright microscope. Digital images were acquired using an Olympus DP72 12.5 Megapixel cooled digital color camera connected to a Dell computer running the Olympus cellSens software platform (Olympus Corporation; https://www.olympus-lifescience.com/ en/software/cellsens/). Image files were then coded and analyzed by an investigator blinded to genotype and saved to a Dell Optiplex 7020 [98].

In Fig 1, semiquantitative analysis of MC1+ cell counts in EC and DG was performed in horizontal brain sections from 16-month EC-Tau/hAPP mice ($n = 6$) and EC-Tau mice ($n = 5$) (Fig 1G and 1H). MC1+ cells within defined ROIs (EC and DG granule cell layers) were counted using the multipoint tool in Fiji and saved via the ROI Manager [99]. The estimated total number of MC1+ neurons/mm$^2$ was then calculated per left hemisphere (EC and DG) in each section and averaged across 3 sections per mouse to generate a representative value for each region. Finally, MC1+ neurons/mm$^2$ values per mouse were compared across genotype for EC and DG. Only neurons with clear somatodendritic accumulation of MC1+ immunoreactivity were included in our analysis. For DAB IHC threshold analyses in Figs 4 and 5, minimum threshold values for 8-bit, grayscale images of immunoperoxidase-stained sections were adjusted interactively under blinded conditions in Fiji using an over/under display mode (red represents pixels above threshold) (S4A Fig and S4D Fig). An ROI was then defined within each image and saved via the ROI Manager. In Fig 4, the total immunoreactivity for 6E10 antibody above threshold was saved as a percentage of total ROI area (mm$^2$) in the hippocampus and used to compare pathological accumulation of hAPP/Aβ in the right-versus-left hippocampus. Mice assigned to control conditions in Fig 4F were as follows: 16-month EC-Tau/hAPP mice with no DREADDs expression or CNO treatment, $n = 2$; 16-month EC-Tau/hAPP mice expressing hM4D$_i$ EC DREADDs but administered a low, chronic dose of CNO (0.5 mg kg$^{-1}$ day$^{-1}$), $n = 4$; 16-month hAPP mice expressing hM4D$_i$ EC DREADDs with no CNO treatment, $n = 3$. In Fig 5, the total immunoreactivity for each tau marker (CP27, AT8 and MC1) above threshold was saved as a percentage of total ROI area (mm$^2$), then a right-versus-left-hemisphere ratio was generated for each region and tissue section analyzed and averaged per mouse. The same minimum threshold value was applied for each pair of images (right-versus-left ROIs) used in our analysis. For 6E10 immunostaining experiments in Fig 5, 3 tissue sections were analyzed per mouse and averaged to generate a representative value reflecting hAPP/Aβ pathology. For CP27, AT8, and MC1 immunostaining experiments in Fig 6, 2 tissue sections were analyzed per marker for each mouse and averaged to generate a representative value reflecting tau pathology.

## Behavioral analysis

Animal performance during in vivo recording sessions was assessed by tracking the position of an infrared LED on the head stage (position sampling frequency, 50 Hz) by means of an overhead video camera. The position data were first centered around the point (0, 0), and then speed-filtered where noted, with only speeds of 5 cm/second or more included in the analysis. Tracking artifacts were removed by deleting samples greater than 100 cm/second and missing positions were interpolated with total durations less than 1 second, and then smoothing the path using a moving average filter with a window size of 300 milliseconds (15 samples on each side). The total distance traveled in the arena (m), % of arena coverage, and average speed (cm/second) during exploration served as dependent measures of interest. For % of arena coverage, the processed position data was plotted and converted to black and white. Using the "bwarea" function in MATLAB, we calculated the total area traveled by the mouse using the binary image and then divided this area by the total area of the arena. For each dependent measure in S2 Fig, individual values per mouse represent the mean of 3 recording sessions. Values for these sessions were also split into the following time-bins and compared across genotype and time-bin: the first 5, 10, and 15 minutes of the total recording session (30 minutes).

## Tools for semiautomated data analysis

*BatchTINTV3*: BatchTINTV3 is a graphical user interface (GUI) written in Python and created as an end-user friendly batch processing solution to complement Axona's command line

interface. BatchTINTV3 sorts the spike data of each session in a chosen directory using Klus-taKwik. The BatchTINTV3 code is freely available and hosted in the following GitHub repository: https://github.com/hussainilab/BatchTINTV3.

**BatchPowerSpectrum.** The percent power values were calculated in MATLAB. A Welch's power spectrum density (PSD) estimate of the LFP was calculated via the "pwelch" function. Using the PSD, the average powers in each of the desired frequency bands were calculated with the "bandpower" function. The average power of each band was then divided by the total power of the signal to produce the percentage power in each of the bands. In situations in which the data were speed filtered (5–30 cm/second), the speed of the animal was calculated and then interpolated so there was a speed value for each LFP value. Then the LFP values were chunked to contain consecutive data points in which the mouse's movements satisfied the minimum and maximum speed requirements. Chunks containing less than 1 second of data were discarded. The aforementioned power calculations were then performed on each of the LFP chunks, and an average of these chunks would yield the final percentage power values for each of the frequency ranges.

**hfoGUI for time-frequency analysis.** A time-frequency representation (TFR) of the LFP was visualized using a GUI written in Python called "hfoGUI.py." The GUI allows for complete control of signal filtering and is equipped with various filter types (butterworth, cheby-shev type 1, chebyshev type 2, etc.), along with the ability to specify filter order, cutoffs, etc. We used third-order butterworth filters in order to maintain consistency with the filter types and order implemented by the Axona data acquisition software/hardware. Specific time windows of the LFP data were selected and subjected to a Stockwell-Transform (s-transform) to visually represent EC-Tau/hAPP and Control genotypes (Fig 3A and 3B). The hfoGUI code is freely available and hosted in the following GitHub repository: https://github.com/hussainilab/hfoGUI.

## Statistical analysis

Statistical analyses were performed in GraphPad Prism 8 and MATLAB. All datasets were tested for normality using the Shapiro–Wilk test. Datasets in which values were not well modeled by a normal distribution were subjected to nonparametric statistical analyses. Unpaired *t*-tests with Welsh's correction were used to perform semiquantitative comparisons of tau pathology in Fig 1. Two-sample Kolmogorov–Smirnov tests were used to compare distributions of average firing rates (AVG FRs) and median ISIs in Fig 2. One-way ANOVAs were used to compare group means of AVG FRs in Fig 2 and the Kruskal–Wallis *H* test was used to compare median ISI values across genotypes. One-way ANOVAs were used to compare the LFP group means in Fig 3. A repeated-measures ANOVA was used to calculate the difference in percentage theta power and total spike counts over time in Fig 4A and 4B (compared with baseline). Paired *t*-tests were used to compare AVG FRs of total EC neurons, NS neurons, and WS neurons at week 6 of CNO treatment to Baseline in Fig 4C–4E. Paired *t*-tests were used to compare 6E10+ immunostaining in the DREADDs-activated right hemisphere versus control left hemisphere in Fig 5. One-sample *t*-tests were used to compare averaged right-versus-left hemisphere ratios for tau markers in each hippocampal ROI to a theoretical mean (1.0) in Fig 6. A 2-sample Kolmogorov–Smirnov test was used to compare the distributions of speed-filtered AVG FRs in S1 Fig. One-way ANOVAs were used to compare group means of AVG FRs and LFP in S1A–S1E Fig, and a Kruskal–Wallis *H* test was used to compare median percentage high-gamma LFP values in S1F Fig. One-way ANOVAs were used to compare the group means in S2B Fig, and linear mixed-effects analyses were performed for each dependent measure in S2C Fig. A repeated-measures 2-way ANOVA with Greenhouse–Geisser correction

was used to calculate the effects of 2 independent variables (drug treatment and time-bin) on normalized spike counts in S3A Fig. A repeated-measures ANOVA was used to calculate the difference in percentage theta power after acute CNO treatment in S3D Fig. Paired *t*-tests were used to compare right-versus-left hippocampal ROI areas in S4B Fig and Pearson's coefficient of determination ($R^2$) are calculated and shown next to scatterplots of right-versus-left ROI areas in S4C Fig and S4E–S4G Fig. Post hoc analyses were performed using Dunnett's or Sidak's multiple comparisons test where noted.

## Supporting information

**S1 Data. Source data for Rodriguez and colleagues (2020)** *PLOS Biology.* The source data used to generate all graphs are included as S1 Data with this manuscript under the file name "Rodriguez et al–PLOS Bio—S1 Data.xlsx." Source data are also available at https://github.com/HussainiLab/PLOS-Biology-manuscript-data. Source data for each main and supplemental figure are arranged by sheet and are labeled (e.g., Fig 1, Fig 2, etc). The following information can be found within each sheet under headers: Mouse ID, Genotype, Sex, and individual data values contributing to the figure. This information is reported on the left side of each sheet (left of a vertical, dark gray bar). The average and/or median values generated per group for statistical comparison are reported on the right side of each sheet. ID, identification. (XLSX)

**S1 Fig. Speed-filtered EC single-unit firing rates and network activity.** A minimum–maximum speed filter was applied to the electrophysiology datasets post hoc to remove single-unit spikes and LFP activity that occurred during bouts of behavioral immobility. For details, please see Materials and methods. *A*. The cumulative frequency distributions of speed-filtered EC neuronal firing rates for each genotype are shown. Distributions from EC-Tau/hAPP ($n$ = 588 cells) and hAPP ($n$ = 532 cells) mice were shifted toward increased firing rates compared with control ($n$ = 386 cells). Two-sample Kolmogorov–Smirnov test: EC-Tau/hAPP versus control: D = 0.268, $p < 0.001$; hAPP versus control: D = 0.248, $p < 0.001$. Insert, speed-filtered firing rates were significantly increased in EC-Tau/hAPP mice and hAPP mice versus Control. One-way ANOVA: $p < 0.001$. Dunnett's multiple comparisons tests, $p < 0.01$. *B-C*. NS and WS cells were examined by genotype after speed filtering. hAPP mice exhibited increased NS neuronal firing rates versus Control. One-way ANOVA: $p < 0.01$. Dunnett's multiple comparisons tests, $p < 0.01$. EC-Tau/hAPP mice exhibited increased WS neuronal firing rates versus control. One-way ANOVA: $p < 0.05$. Dunnett's multiple comparisons tests, $p < 0.05$. *D*. Speed-filtered percentage theta power values were significantly reduced in EC-Tau/hAPP mice compared to Control. One-way ANOVA: $p < 0.05$. Dunnett's multiple comparisons tests, $p < 0.05$. *E*. Speed-filtered percentage low-gamma power values were significantly increased in EC-Tau/hAPP and hAPP mice versus control. One-way ANOVA: $p < 0.001$. Dunnett's multiple comparisons tests, $p < 0.01$. *F*. No differences were detected across genotype in speed-filtered percentage high gamma power. Bar graphs represent mean ± SEM. Individual values per mouse appear as overlays. $^*p < 0.05$; $^{**}p < 0.01$; $^{***}p < 0.001$. Source data are available in S1 Data. EC, entorhinal cortex; hAPP, human amyloid precursor protein; LFP, local field potential; NS, narrow-spiking; SEM, standard error mean; WS, wide-spiking. (PDF)

**S2 Fig. hAPP/Aβ and tau pathology does not affect motivated foraging behavior in an open field.** Locomotor activity was assessed in vivo by analyzing the position data of each mouse during active exploration in an open field. Sample sizes are as follows: Control, $n$ = 10 total: $n$ = 6 female, $n$ = 4 male; EC-Tau, $n$ = 8 total: $n$ = 4 female, $n$ = 4 male; hAPP, $n$ = 8 total:

$n$ = 4 female, $n$ = 4 male; EC-Tau/hAPP, $n$ = 10 total: $n$ = 6 female, $n$ = 4 male. **A**. Representative trajectories are shown for one recording session per genotype. Scale bar, 20cm. **B**. The following parameters in the open field were analyzed and compared across genotype: the total distance traveled (m), the % arena coverage and average speed (cm/second). No significant group differences were detected on any measure. One-way ANOVA tests: Total distance (meter), $F_{(3,32)}$ = 0.228, $p$ > 0.05; % of arena coverage, $F_{(3,32)}$ = 0.345, $p$ > 0.05; average speed (cm/second), $F_{(3,32)}$ = 0.237, $p$ > 0.05. **C**. No significant group differences in behavioral measures were detected in the first 5-, 10-, or 15-minute time-bins of the recording sessions. Linear mixed-effects analyses were performed for each dependent measure. Bar graphs represent mean ± SEM. Individual mouse values appear as overlays and represent the mean of 3 averaged recording sessions per mouse. Source data are available in S1 Data. Representative Axona.pos files for each mouse are available at: https://github.com/HussainiLab/PLOS-Biology-manuscript-data. Aβ, amyloid-beta; EC, entorhinal cortex; hAPP, human amyloid precursor protein; SEM, standard error mean.
(PDF)

**S3 Fig. Acute EC DREADDs activation in mouse models of AD pathology.** Single injections of CNO (i.p.) were used to determine salient recording measures of EC DREADDs activation in vivo. Long-term recordings were performed in 12- to 16-month mice to first determine the onset and duration of altered single-unit activity after hM4D$_i$ EC DREADDs activation. **A.** Averaged total spike counts are shown for 60 min recordings following 5–10 mg/kg CNO and Saline injections ($n$ = 6 mice: $n$ = 1 female, $n$ = 5 male). Spike counts were first normalized to individual mouse baseline measures collected prior to drug treatment and then plotted in 5-minute time-bins for 60 minutes total. hM4D$_i$ EC DREADDs activation begins to affect total spike counts approximately 20 minutes after injection, with significant post hoc comparisons evident at 40 minutes ($p$ < 0.05), 50 minutes ($p$ < 0.01), and 55 minutes ($p$ < 0.05) time-bins versus Saline. Repeated-measures 2-way ANOVA with Greenhouse–Geisser correction: (time × drug treatment) $F_{(11,110)}$ = 5.448, $p$ < 0.001. (Time) $F_{(2.4,23.5)}$ = 3.112, $p$ >0.05. (Drug Treatment) $F_{(1,10)}$ = 30.490, $p$ <0.001. Sidak's multiple comparisons test. Diamond (purple), hM4D$_i$ EC DREADDs; circle (white), aaline. **B.** Total spike counts (normalized to baseline) following hM4D$_i$ EC DREADDs activation are shown for one 4-hour recording session in a 16-month hAPP mouse. Mean spiking activity in 15 min time-bins in shown. For 1- to 4-hour time points, the last 15 minutes of each hour was analyzed. **C**. Acute hM4D$_i$ EC DREADDs activation measurably impacted percentage theta power in vivo. Representative LFP traces are shown for both CNO and saline conditions. Notable differences in EC theta modulation are shown after 10 mg/kg CNO treatment. **D**. hM4D$_i$ EC DREADDs activation decreased % theta power values compared to baseline, with a significant reduction detected at 45–60 minutes post-CNO ($p$ < 0.05) ($n$ = 8 mice: $n$ = 4 female, $n$ = 4 male). Repeated-measures ANOVA: $F_{(2,7)}$ = 7.079, $p$ < 0.05. Dunnett's multiple comparisons test: 45–60 minutes versus baseline, $p$ < 0.05. Mean percentage theta power in 15-minute time-bins in shown. Colored overlays represent individual percentage theta values per mouse over 3 time points. $^*p$ < 0.05; $^{**}p$ < 0.01. Source data are available in S1 Data. AD, Alzheimer's disease; CNO, clozapine-n-oxide; DREADD, designer receptor exclusively activated by a designer drug; EC, entorhinal cortex; hM4D$_i$, human M4 DREADD receptor; i.p., intraperitoneal; LFP, local field potential.
(PDF)

**S4 Fig. Right-versus-left hemisphere ROI measures for DAB IHC analysis.** Area measurements of hippocampal ROI downstream from hM4D$_i$ DREADDs-expressing EC (right hemisphere) were compared to contralateral hippocampal ROIs (left hemisphere). **A**. *Top*, 8-bit gray scale images of 6E10+ immunoreactivity in a horizontal brain section from a 16-month

EC-Tau/hAPP mouse after chronic hM4D$_i$ EC DREADDs activation. A minimum threshold value was first applied to each image and then the ROIs were defined for right and left hippocampus (magenta). The % area of 6E10+ immunoreactivity above threshold was used to quantify hAPP/Aβ accumulation within the right and left hippocampal ROIs. Scale bars, 500μm. *Bottom*, Higher magnification of 6E10+ immunoreactivity within ROIs. Black pixels depict 6E10+ immunoreactivity above threshold on a white background. Scale bars, 250μm. ***B.*** *Left*, no differences were detected in right-versus-left hippocampal ROIs (mm$^2$) in EC-Tau/hAPP mice after chronic hM4D$_i$ EC DREADDs activation. Paired *t*-test: $t(8) = 1.422$, $p > 0.05$. Right, no differences were detected in right-versus-left hippocampal ROIs in mice subjected to control conditions. Paired *t*-test: $t(8) = 1.856$, $p > 0.05$. Individual values represent the average of 3 sections per mouse and appear as colored bar overlays. Mean ROI area is depicted within bar graphs. ***C***. A scatter plot of right-versus-left hippocampal ROIs (mm$^2$) is shown for 6E10 + immunostained brain sections (total, $n = 18$ data coordinates; $n = 3$ sections/mouse; $n = 18$ mice). The coefficient of determination ($R^2 = 0.7593$) is shown below the scatter. ***D***. High-magnification images were taken at 20× for semiquantitative analysis of tau immunoreactivity within hippocampal subregions. MC1+ staining is shown for 1 EC-Tau/hAPP mouse after chronic hM4D$_i$ EC DREADDs activation. Top panel, 8-bit grayscale images of MC1+ immunoreactivity are shown for 3 hippocampal regions analyzed: DG, CA1, and the Sub. MC1+ immunoreactivity within the defined ROI (black outline) appears as red pixels above minimum threshold. Bottom panel, accompanying black and white over/under images are shown to emphasize MC1+ signal in ROIs, 100 μm. ***E-G.*** Scatter plots of right-versus-left ROIs are shown for each hippocampal region analyzed with their respective coefficient of determination. DG: $R^2 = 0.4985$; CA1: $R^2 = 0.8747$; Sub: $R^2, = 0.8716$. Individual values represent the average ROI size derived from 2 sections per antibody per mouse (Total, $n = 18$ data coordinates; $n = 6$ sections/mouse; $n = 6$ mice). Source data are available in S1 Data. AD, Alzheimer's disease; CA1, Cornu Ammonis 1; CNO, clozapine-n-oxide; DAB, diaminobenzidine; DG, dentate gyrus; DREADD, designer receptor exclusively activated by a designer drug; EC, entorhinal cortex; hAPP, human amyloid precursor protein; hM4Di, human M4 DREADD receptor; IHC, immunohistochemistry; LFP, local field potential, MC1, antibody for misfolded tau; ROI, region of interest; Sub, subiculum.
(PDF)

## Acknowledgments

We thank Chukwuma Onyebuenyi, BS, and Paula Choconta, BS, for expert technical assistance with electrophysiological data analysis and image processing. We thank Helen Y. Figueroa, BS, for maintaining the EC-Tau/hAPP mouse colonies. We thank Dr. Peter Davies for generously providing tau antibodies. The authors would also like to thank Drs. Wai Haung Yu, Catherine Clelland, Natura Myeku, and Tal Nuriel for helpful discussions and comments regarding the manuscript.

## Author Contributions

**Conceptualization:** Gustavo A. Rodriguez, Karen E. Duff, S. Abid Hussaini.

**Data curation:** Gustavo A. Rodriguez, Geoffrey M. Barrett.

**Formal analysis:** Gustavo A. Rodriguez, Geoffrey M. Barrett.

**Funding acquisition:** Gustavo A. Rodriguez, Karen E. Duff, S. Abid Hussaini.

**Investigation:** Gustavo A. Rodriguez, Karen E. Duff.

**Methodology:** Gustavo A. Rodriguez.

**Project administration:** Gustavo A. Rodriguez, Karen E. Duff, S. Abid Hussaini.

**Resources:** Gustavo A. Rodriguez, Karen E. Duff, S. Abid Hussaini.

**Software:** Geoffrey M. Barrett.

**Supervision:** Gustavo A. Rodriguez, Karen E. Duff, S. Abid Hussaini.

**Validation:** Gustavo A. Rodriguez.

**Visualization:** Gustavo A. Rodriguez.

**Writing – original draft:** Gustavo A. Rodriguez, Karen E. Duff, S. Abid Hussaini.

**Writing – review & editing:** Gustavo A. Rodriguez, Geoffrey M. Barrett, Karen E. Duff, S. Abid Hussaini.

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
