## [Editor Report · Decision Letter 0]

9 Mar 2020

Dear Abid, 

Thank you for submitting your revised manuscript entitled "Chemogenetic attenuation of neuronal activity in the entorhinal cortex reduces Aβ and tau pathology in the hippocampus" for consideration as a Research Article by PLOS Biology.

Your revision has now been evaluated by the PLOS Biology editorial staff, as well as by the original Academic Editor, and I am writing to let you know that we would like to send your submission out for external peer review.

Please re-submit your manuscript within two working days, i.e. by Mar 11 2020 11:59PM.

Kind regards,

Gabriel Gasque, Ph.D.,

Senior Editor

PLOS Biology

---

## [Decision Letter · Decision Letter 1]

18 Apr 2020

Dear Abid,

Thank you very much for submitting a revised version of your manuscript "Chemogenetic attenuation of neuronal activity in the entorhinal cortex reduces Aβ and tau pathology in the hippocampus" for consideration as a Research Article at PLOS Biology. This revised version of your manuscript has been evaluated by the PLOS Biology editors and by the original Academic Editor. As mentioned previously, only one of the original reviewers, reviewer 1, agreed to re-review. Thus, we sought advice from another three independent experts: reviewers 5, 6, and 7. 

In light of the reviews (below), we are pleased to offer you the opportunity to address the comments from reviewers 5 and 6 in a revised version that we anticipate should not take you very long. We will then assess your revised manuscript and your response to the reviewers' comments, and we may consult the reviewers again.

We expect to receive your revised manuscript within 1 month.

Please email us (plosbiology@plos.org) if you have any questions or concerns or would like to request an extension. At this stage, your manuscript remains formally under active consideration at our journal; please notify us by email if you do not intend to submit a revision so that we may end consideration of the manuscript at PLOS Biology.

**IMPORTANT - SUBMITTING YOUR REVISION**

Your revisions should address the specific points made by reviewers 5 and 6. Please submit the following files along with your revised manuscript:

*Resubmission Checklist*

*Published Peer Review*

*PLOS Data Policy*

*Blot and Gel Data Policy*

Sincerely,

Gabriel Gasque, Ph.D., 

Senior Editor

PLOS Biology

REVIEWS:

Reviewer #1: I am satisfied with the corrections and changes to the manuscript. No further comments.

Reviewer #2: [did not re-review]

Reviewer #3: [did not re-review]

Reviewer #4: [did not re-review]

Reviewer #5 (new reviewer): In this manuscript, authors Rodriguez et al. investigate pathology and network activity in the entorhinal cortex (EC) and downstream hippocampal (HP) pathway in a novel transgenic model overexpressing mutant human APP and EC-specific Tau. Their results suggest that amyloid drives hyperexcitability and the propagation of tau pathology to the HP. Following this logic, the authors chemogenetically decreased neuronal activity in the EC and showed that that this manipulation reduced hyperphosphorylated and abnormally conformed tau in some downstream regions. This work provides interesting data on the interplay between A� and tau pathology in Alzheimer's disease. Overall the paper is quite good, and the authors are to be commended for the new data and extensive revisions they added in response to the first set of reviews. Addressing a few additional points could further improve the impact of this work. 

1. Figure 1 shows amyloid and tau pathology at multiple time points in the EC-Tau/hAPP cross, as well as in the EC-Tau mice at 16 months. It would be helpful to also show results from the hAPP transgenic mice.

2. The box legend in Figure 2 illustrating the genotypes is difficult to locate, and should be repositioned closer to 2B.

3. The CP27 staining in Figure 5A-E suggests that the tau pathology that "spread" from the EC to the hippocampus was composed of mutant human tau. Can the authors comment on whether corrupted endogenous mouse tau was also recruited?

4. In line 220, it is unclear what "Single, high dose CNO (5 and 10mg/kg, i.p.) injections reliably induced DREADDs expression" means. The expression of DREADDs is independent of ligand administration. In the response to reviewers, it was stated that this sentence had been changed, but it still reads the same in the revised version I received.

5. Figure 4 illustrates a reduction in spiking. It is not clear whether this reduction was seen in NS (inhibitory) or WS (excitatory) neurons, or both. It would be helpful to show data separated by neuron type. Finally, was the reduction in 6E10+ pathology also seen in the EC, or just the hippocampus?

6. The authors should be conservative about generalizing their data, since they only examined particular mutations and one brain region, and there are different ways of measuring neuronal activity. There are reports that tau can cause hyperexcitability, even though it was not evident in this study. For example, how do the authors think about their data in the context of Sohn et al., Neuron 104:458-70? How do the authors think about their data given that tau pathology in human Alzheimer's disease is composed of wild-type tau, not P301L mutant tau? Some discussion of these issues should be added to the manuscript.

Reviewer #6 (new reviewer, also as an attachment): In this manuscript, Rodriguez et al. use a new transgenic mouse line expressing both Tau and hAPP in the entorhinal cortex (EC) to show that hAPP aggravates tau aggregation in the entorhinal cortex and accelerates pathological tau spread into the hippocampus. The authors identify hAPP/A�, and not tau, as the major trigger for neuronal hyperactivity and impaired theta in the EC. Finally, the authors use a chemogenetic approach to attenuate neuronal hyperactivity in the EC and they provide evidence for reduced hAPP/Aβ. accumulation and reduced pathological tau spread downstream in the hippocampus. Overall this study nicely complements previous studies showing abnormal activity patterns in the context of AD, and its link with disease progression.

It is a revision of a manuscript for which the main criticisms dealt with small sample size and with quantification methods. The authors have made the appropriate efforts to answer to most of the concerns of the reviewers. They have increased the sample size when necessary, and improved the presentation. The manuscript should now be potentially suitable for publication, provided minor revisions indicated below:

Figure 1A - It is quite unfortunate that there is no quantification of the plaques or hAPP/A� immunoreactivity, in the different areas and models as there is for MC1 immunoreactivity. See line 241, this has been done in part of the CNO experiments.

Figure 1D - What is the use of these panels, and what differences with middle panels in 1A? In addition, what is the meaning and utility of the thick black lines? 

Figure 4 and 5. The effect of chemogenetic attenuation of EC neuronal activity on A� and Tau pathology is perhaps the most original result of the paper. The effect occurs on 16 months-old EC-Tau/hAPP mice which already exhibit significant level of pathology. A treatment with CNO for 6 weeks diminishes the level of both amyloid (Fig4D, E) and Tau (Fig5J, O). Thus, CNO does not inhibit the appearance of AD-related hallmarks but rather seems to partially revert the pathology. Surprisingly, there is no mention about the kinetic of the events and about the possible mechanisms involved: inhibition of Ab production and Tau phosphorylation, versus clearance of plaques and tangles. At minimum, this should be addressed in the discussion.

Line 226 - The authors should provide a comparison between the levels of neuronal activity (and theta power) attained after CNO treatment, and those observed in WT mice at the same age. Does the CNO treatment lead to normalization of neuronal activity? Or hypoactivity? In both NS and WS cell types?

Line 275. "CP27+ immunostaining appeared to be selectively decreased". Why is there no quantification? These sentences are unclear, what differences should be shown in Fig 5C,D?

Line 386 - " blunted hyperactivity in NS neurons of EC-Tau/hAPP mice (Figure 2E), which may represent an early, synergistic effect of tau on Aβ-associated inhibitory interneuron dysfunction that could precede subsequent impairments in excitatory neurons and gross network function". Indeed, the authors should suggest explanations for the blunted hyperactivity of putative interneurons in EC-Tau/hAPP mice. But I must say that the sentence above is quite unclear, and should at least be rephrased, and possibly split in several sentences.

Reviewer #7 (new reviewer): Rodriguez and collaborators propose a convincing report that focuses on the functional relationship between co-expression of human APP/Tau and tau pathology in the EC-Hippocampal network. The chemogenetic approaches are undoubtedly fascinating methods to decrease neuronal activity and observe the resulting effects on abeta/tau pathology.

This Reviewer has appreciated the Author's efforts to ameliorating the manuscript and to adequately reply Reviewers requests.

Thus, this Reviewer recommends the Ms for publication.

---

## [Decision Letter · Decision Letter 2]

25 Jun 2020

Dear Abid,

Thank you for submitting your revised Research Article entitled "Chemogenetic attenuation of neuronal activity in the entorhinal cortex reduces Aβ and tau pathology in the hippocampus" for publication in PLOS Biology. I have now obtained advice from the original reviewers 5 and 6, and have discussed their comments with the Academic Editor. 

We're delighted to let you know that we're now editorially satisfied with your manuscript. However before we can formally accept your paper and consider it "in press", we also need to ensure that your article conforms to our guidelines. A member of our team will be in touch shortly with a set of requests. As we can't proceed until these requirements are met, your swift response will help prevent delays to publication. Please also make sure to address the data and other policy-related requests noted at the end of this email.

*Copyediting*

*Published Peer Review History*

*Early Version*

*Submitting Your Revision*

Sincerely,

Gabriel Gasque, Ph.D., 

Senior Editor

PLOS Biology

DATA POLICY:

>> Please check the labels and/or data for Fig 5 in S1 Data. 

>> Data for Fig S4C seem to be missing in S1 Data.

>> The link to https://github.com/HussainiLab/PLOS-Biology-manuscript-data seems broken

>> Please ensure that figure legends in your manuscript include information on where the underlying data can be found and that your supplemental data file/s has a legend.

>> Please ensure that your Data Statement in the submission system accurately describes where your data can be found.

Reviewer remarks:

Reviewer #5: The authors have adequately addressed all of my comments.

Reviewer #6: The authors have carefully taken into consideration the concerns raised. This is a revised paper which should now be accepted for publication.

---

## [Editor Report · Decision Letter 3]

4 Aug 2020

Dear Dr Hussaini,

On behalf of my colleagues and the Academic Editor, Gillian P. Bates, I am pleased to inform you that we will be delighted to publish your Research Article in PLOS Biology. 

Early Version

PRESS 

Kind regards,

Vita Usova

Publication Editor, 

PLOS Biology

on behalf of

Gabriel Gasque,

Senior Editor

PLOS Biology